# MICOS coordinates with respiratory complexes and lipids to establish mitochondrial inner membrane architecture

Jonathan R Friedman[1], Arnaud Mourier[2], Justin Yamada[1], J Michael McCaffery[3], Jodi Nunnari[1]*

[1]Department of Molecular and Cellular Biology, College of Biological Sciences, University of California, Davis, Davis, United States; [2]Department of Mitochondrial Biology, Max Planck Institute for Biology of Ageing, Cologne, Germany; [3]Integrated Imaging Center, Johns Hopkins University, Baltimore, United States

**Abstract** The conserved MICOS complex functions as a primary determinant of mitochondrial inner membrane structure. We address the organization and functional roles of MICOS and identify two independent MICOS subcomplexes: Mic27/Mic10/Mic12, whose assembly is dependent on respiratory complexes and the mitochondrial lipid cardiolipin, and Mic60/Mic19, which assembles independent of these factors. Our data suggest that MICOS subcomplexes independently localize to cristae junctions and are connected via Mic19, which functions to regulate subcomplex distribution, and thus, potentially also cristae junction copy number. MICOS subunits have non-redundant functions as the absence of both MICOS subcomplexes results in more severe morphological and respiratory growth defects than deletion of single MICOS subunits or subcomplexes. Mitochondrial defects resulting from MICOS loss are caused by misdistribution of respiratory complexes in the inner membrane. Together, our data are consistent with a model where MICOS, mitochondrial lipids and respiratory complexes coordinately build a functional and correctly shaped mitochondrial inner membrane.

*For correspondence: jmnunnari@ucdavis.edu

## Introduction

Mitochondria are double membrane-bound organelles whose shape is instrumental to their function in diverse cellular processes such as metabolism and apoptosis (*Labbe et al., 2014*). The inner mitochondrial membrane has the highest density of protein in the cell and is differentiated into three distinct interconnected domains: the boundary region, which are flattened membranes that lie in close apposition to the outer membrane; cristae membranes, which are lamellar invaginations with highly curved edges; and cristae junctions, which are relatively narrow tubules that connect cristae to the boundary membrane and may act as a physical partitioning mechanism that prevents and/or regulates the intermixing of proteins between cristae and the boundary domains (*Mannella et al., 1994*). Compositionally, the boundary membrane is enriched for import and assembly machinery for the mitochondrial proteome (*Vogel et al., 2006*; *Wurm and Jakobs, 2006*). In contrast, cristae membranes house assembled electron transport chain protein complexes and ATP synthase, which function together to synthesize ATP via oxidative phosphorylation. Specific combinations of electron transport chain complexes further assemble into large mega-Dalton supercomplexes in cristae in a manner dependent on the mitochondrial lipid cardiolipin and act to facilitate electron transport, and likely as diffusion traps to promote their sorting into cristae (*Cruciat et al., 2000*; *Zhang et al., 2002*;

**eLife digest** Structures called mitochondria provide energy that cells need to live and grow. To do this, mitochondria convert energy stored within sugars and other carbon-rich compounds into the energy currency of cells, a molecule called adenosine triphosphate (called ATP for short). Defective mitochondria can cause cells to starve and also cause severe human diseases.

A double membrane surrounds each mitochondrion. The outer membrane allows proteins and other substances to enter, while the inner membrane is elaborately folded and contains several groups of proteins—or complexes—including the respiratory complexes that generate ATP. Proper inner membrane folding is critically important. The membrane folds are held in place by structures called cristae junctions, which may also help to restrict proteins to particular areas of the inner membrane.

A large inner membrane complex of proteins known as MICOS is important for organizing the inner membrane into folds, although exactly how it does so is not fully understood. MICOS consists of at least six different proteins, most of which are found across yeast and animal species. Friedman et al. have now analyzed how the MICOS complex assembles on the inner membrane in yeast cells using a combination of fluorescence and electron microscopy, proteomics and biochemistry. This revealed that in yeast, MICOS is made up of two independent sub-complexes bridged together by a protein called Mic19, which additional experiments suggest controls the number and positions of the cristae junctions that hold the folds of the inner membrane in place.

As part of the approach to understand MICOS complex organization, Friedman et al. removed the six MICOS proteins from yeast cells. Inside these cells, the inner mitochondrial membrane was misfolded. Furthermore, the respiratory complexes did not work normally and as a consequence the cells were unable to grow normally, suggesting that the correct distribution of respiratory complexes in the inner membrane is important for ATP production and depends on MICOS.

These results indicate that MICOS stabilizes the structure of the inner membrane and organizes it into an efficient energy-generating machine. In many human mitochondrial diseases, the inner membrane of mitochondria folds incorrectly, in similar ways to the misfolding seen in the yeast cells that did not contain the MICOS complex. Therefore, the MICOS complex may also influence how these diseases develop.

*Pfeiffer et al., 2003*; *Acehan et al., 2011*; *Lapuente-Brun et al., 2013*; *Wilkens et al., 2013*). ATP synthase also assembles in a regulated manner into dimers and dimer oligomers, whose unique structure generates and/or stabilizes the high curvature at the edges of lamellar cristae (*Abrahams et al., 1994*; *Paumard et al., 2002*; *Strauss et al., 2008*).

The proper differentiation of the mitochondrial inner membrane into separate domains is critical for mitochondrial function. Indeed, a hallmark feature of mitochondrial diseases is aberrant mitochondrial ultrastructure and shape (*Zick et al., 2009*). Inner membrane structure is also a critical feature of apoptosis, where cristae junctions are remodeled to facilitate the release of the cell death mediator, cytochrome *c* (*Scorrano et al., 2002*). Internal mitochondrial architecture may also interface with the external mitochondrial environment, including contacts with the endoplasmic reticulum (ER) as nucleoids associate with both mitochondrial cristae and mitochondrial division sites, which are marked by sites of ER contact (*Brown et al., 2011*; *Kopek et al., 2012*; *Murley et al., 2013*). Thus, proper mitochondrial ultrastructure is critical for a multitude of mitochondrial and cellular functions.

The mechanisms by which inner membrane domains are established and maintained are poorly understood. In addition to respiratory supercomplexes, ATP synthase oligomers and mitochondrial lipid composition, the inner membrane fusion dynamin, Mgm1/OPA1, and scaffolding proteins such as prohibitins have been proposed to play roles in inner membrane structure (*Frezza et al., 2006*; *Meeusen et al., 2006*; *Merkwirth et al., 2008*). These factors are interdependent—for example, cardiolipin is required for both respiratory supercomplex assembly and Mgm1/OPA1 self-assembly and function and the prohibitins are required to maintain normal mitochondrial lipid homeostasis (*DeVay et al., 2009*; *Osman et al., 2009*). This interdependency suggests that inner membrane differentiation is a highly cooperative process, however, exactly how these determinants work

together to correctly shape and organize the mitochondrial membrane to ultimately lead to proper respiratory function is also not understood.

The recently identified MICOS complex (previously named MitOS or MINOS) has been proposed to act as a master regulator/integrator of mitochondrial inner membrane shape and organization (*Harner et al., 2011*; *Hoppins et al., 2011*; *von der Malsburg et al., 2011*; *Alkhaja et al., 2012*; *Pfanner et al., 2014*). Consistently, MICOS interacts both physically and functionally with cardiolipin, import machinery, and respiratory complexes (*Rabl et al., 2009*; *Hoppins et al., 2011*; *Bohnert et al., 2012*; *Korner et al., 2012*; *Zerbes et al., 2012*; *Weber et al., 2013*; *Harner et al., 2014*). The MICOS complex is also embedded in the inner membrane with domains facing the intermembrane space that mediate the formation of heterologous structures localized to the inner boundary membrane (*Hoppins et al., 2011*). It is comprised of six core subunits in yeast: Mic60, Mic10, Mic19, Mic27, Mic26, and Mic12, that, with the exception of Mic12, have mammalian homologs (*Xie et al., 2007*; *Mun et al., 2010*; *Darshi et al., 2011*; *Head et al., 2011*; *Alkhaja et al., 2012*; *An et al., 2012*; *Weber et al., 2013*). Single MICOS subunit deletion causes a characteristic mitochondrial inner membrane morphological defect in cells, consisting of extended, stacked, lamellar inner membranes and a reduction of the number of cristae junctions, with a consequent lamellar mitochondrial shape defect. The common cellular phenotypes of single MICOS subunit deletions indicate that they perform a shared function. However, expression analysis indicates that Mic60 and Mic10 function uniquely as 'core components' that direct a hierarchal MICOS assembly as *MIC60* deletion causes Mic19 instability and *MIC10* deletion causes Mic27 instability (*Harner et al., 2011*; *Hoppins et al., 2011*; *von der Malsburg et al., 2011*). In addition, specific pairwise combinations of MICOS subunit deletions can produce either positive or negative genetic interactions (*Hoppins et al., 2011*), indicating that although MICOS subunits act cooperatively, they also perform non-redundant roles within mitochondria. For example, Mic60 plays a role in import independently of the MICOS complex (*von der Malsburg et al., 2011*).

Here we examine the role of MICOS in the lateral organization of the mitochondrial inner membrane by examining how MICOS is assembled and how MICOS cooperates with the surrounding mitochondrial environment. Using yeast cells deficient for all six components of MICOS, we identify two major MICOS organizing centers, Mic60 and Mic27/Mic10/Mic12. Our data indicate that Mic27/Mic10/Mic12 assembles at cristae junctions in a respiratory complex- and cardiolipin-dependent manner. In contrast, Mic60 assembles and organizes independently of other MICOS subunits, cardiolipin, and the respiratory machinery, suggesting that Mic60 assemblies may intrinsically mark nascent cristae junctions. MICOS subcomplexes are bridged together by Mic19, which our data indicates controls the copy number and position of cristae junctions within mitochondria. Together, our findings demonstrate how the MICOS complex works with respiratory complexes and the mitochondria lipid environment to establish inner membrane architecture, organization and function.

## Results

### The MICOS complex is required for oxidative phosphorylation and normal mitochondrial ultrastructure and morphology

To address the functional roles and organization of MICOS, we generated a yeast strain lacking all core MICOS components using a Cre-*lox* recombination system, which allowed for sequential MICOS subunit gene deletion and selection marker rescue (*Guldener et al., 1996*). Using this approach, we constructed ΔMICOS—a strain containing 'clean,' unmarked deletions of all six MICOS genes. In contrast to single MICOS subunit deletions, ΔMICOS cells displayed a severe respiratory growth defect, as assessed by growth on the non-fermentable carbon source, glycerol (*Figure 1A*, right panel). Importantly, reintroduction of four of six MICOS genes at their native loci complemented the growth defect, validating the ΔMICOS strain (*Figure 1A*). The mitochondrial morphology defect of the ΔMICOS strain, as assessed by fluorescence microscopy using the matrix-targeted fluorescent protein, mito-DsRed, was also more penetrant as compared to mitochondria in individual MICOS deletion cells (*Figure 1B*). Although the predominant lamellar shape phenotype present in single MICOS deletion mutants was observed (56% of Δ*mic60* cells vs 21% of ΔMICOS cells), a majority of ΔMICOS cells also had bulbous mitochondrial structures with a beads on a string appearance, and hollow, large spherical structures that were sometimes continuous with the bulbous structures

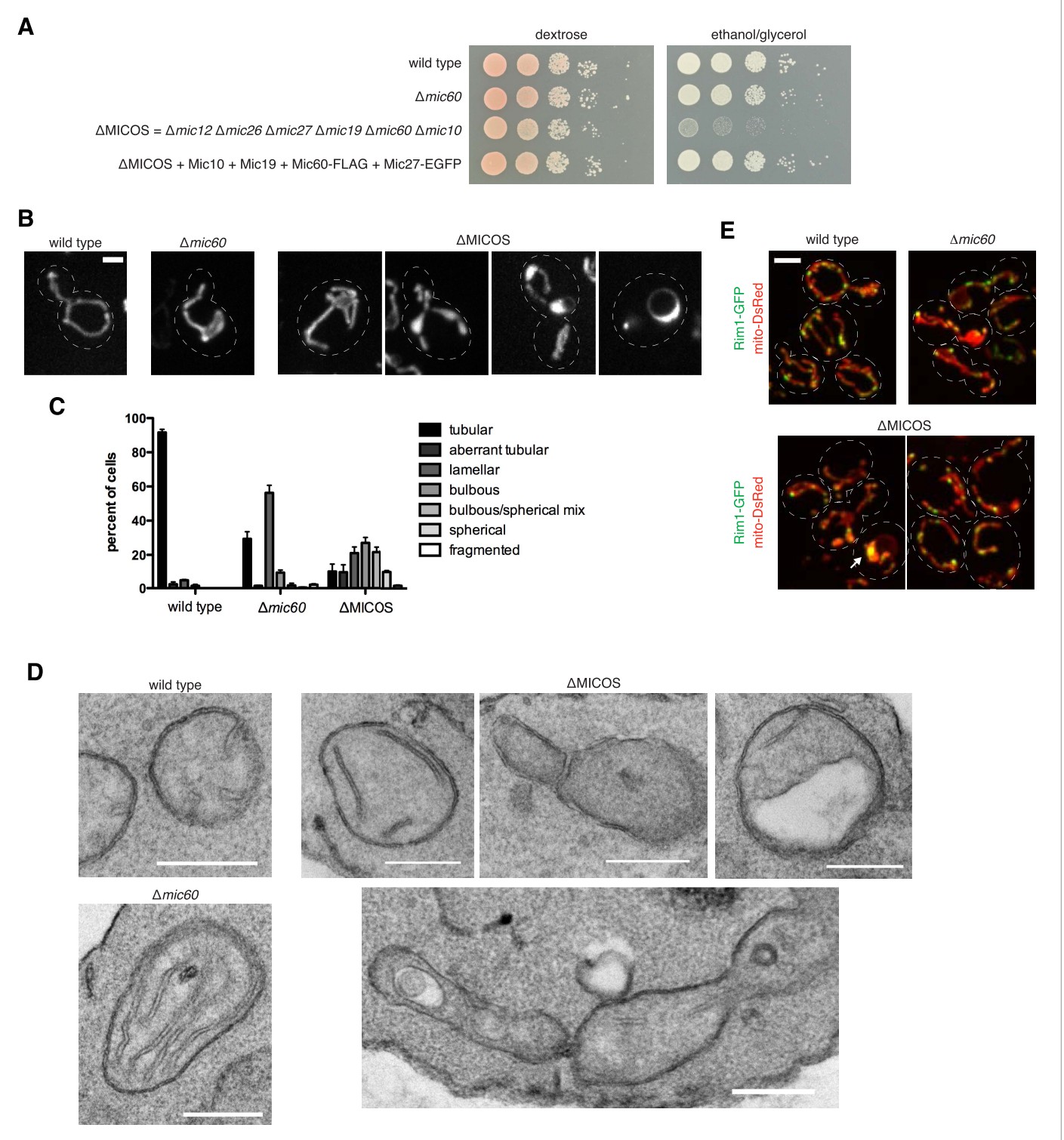

**Figure 1**. The MICOS complex is required for oxidative phosphorylation and normal mitochondrial ultrastructure and morphology. (**A**) Serial dilutions of the indicated yeast cells were plated on media containing glucose (left) and the non-fermentable carbon source, glyercol (right). (**B**) Mitochondrial morphology in the indicated strains was determined by imaging cells expressing the matrix marker mito-dsRed. Z-projections of confocal fluorescence images are shown, except for the right panel of ΔMICOS, which is a single plane. (**C**) Quantification of mitochondrial morphologies from cells imaged as in (**B**) were categorized. Approximately 100 cells from three independent experiments were quantified and data are represented as mean ± SEM. (**D**) Representative electron microscopy images are shown of chemically fixed yeast cells from the indicated strains. (**E**) Confocal fluorescence microscopy z-projections of cells from the indicated strains expressing mito-DsRed and the functional nucleoid marker Rim1-GFP are shown. The arrow marks

Figure 1. Continued

aggregation of Rim1-GFP in a ΔMICOS cell. Scale bars: (**B**) 2 µm; (**D**) 500 nm; (**E**) 3 µm. See also *Figure 1—figure supplement 1*.

The following figure supplement is available for figure 1:

**Figure supplement 1**. Rim1 is a functional marker of nucleoids.

(11% of Δ*mic60* cells vs 59% of ΔMICOS cells) (*Figure 1C*). These spherical structures are consistent with what have previously observed in a relatively small fraction of Δ*mic60* cells (*Itoh et al., 2013*). EM analysis of ΔMICOS cells revealed mitochondrial features analogous to those observed by fluorescence microscopy, including extended lamellar cristae, septated inner membranes, mitochondria with larger diameter structures, and hollow mitochondria, respectively (*Figure 1D*).

We examined the distribution of mitochondrial genomes by imaging cells expressing a tagged mitochondrial DNA-binding protein, Rim1-GFP, a functional marker of nucleoids (*Van Dyck et al., 1992*) (*Figure 1—figure supplement 1*). In wild type and Δ*mic60* cells, genomes were relatively uniformly distributed throughout mitochondria, however in Δ*mic60* cells, nucleoids localized to the edge of lamellar mitochondrial regions, suggesting they may be excluded from regions of stacked, lamellar inner membranes (*Figure 1E*). In ΔMICOS cells, while Rim1-marked nucleoids were maintained, a fraction of nucleoids appeared aggregated as compared to in wild type cells (marked by an arrow in *Figure 1E*). Aggregated nucleoids were mostly observed in spherical mitochondria in ΔMICOS cells, a phenotype previously observed in a relatively smaller fraction of Δ*mic60* and Δ*mic10* cells (*Itoh et al., 2013*). Our characterization of ΔMICOS cells indicates that, although MICOS subunits have non-redundant functions, they act together to buffer the loss of function caused by deletion of single components. However, these observations point to the critical role of the intact MICOS complex in the maintenance of mitochondrial function, morphology, and distribution of mitochondrial genomes in cells.

## Mic60 assembles independently of other MICOS members and Mic19 regulates its distribution within mitochondria

Members of the MICOS complex are localized in a non-uniform focal-like pattern along the inner membrane in both yeast and mammalian cells (*Hoppins et al., 2011*; *Jans et al., 2013*) (*Figure 2A*). These substructures have been postulated to reflect a scaffold-like function of the complex at cristae junctions (*Hoppins et al., 2011*). To gain insight into the molecular basis and functional significance of MICOS organization, we utilized the ΔMICOS strain as a tool to examine the formation of mitochondrial substrctures using functional fluorescent protein tagged versions of MICOS components re-integrated at their respective loci. With the exception of Mic10, all C-terminal fluorescent protein-tagged MICOS subunits were functional based on their ability to maintain wild type-like tubular mitochondria in a wild type cell background (*Figure 2A*). In contrast to wild type cells, in ΔMICOS cells the distribution observed for MICOS components Mic27, Mic19, Mic12 and Mic26 was uniform, similar to the matrix marker (*Figure 2B*). Mic60, however, uniquely localized to discrete foci in a majority of ΔMICOS cells (*Figure 2B,F*), indicating that Mic60 localizes to substructures independent of its interaction with the MICOS complex, suggesting that Mic60 may have an intrinsic ability to self-assemble.

To test this possibility, we performed proteomic analysis of ΔMICOS cells expressing both FLAG-tagged Mic60 at its endogenous locus and EGFP-tagged Mic60 integrated at a second locus driven by its native promoter (pMic60-EGFP). The mild overexpression of Mic60 as a consequence of two genomic copies did not interfere with the ability of pMic60-EGFP to localize to foci (*Figure 2—figure supplement 1*). Purification of Mic60 from cross-linked extract from these cells using either α-FLAG or α-GFP antibodies specifically co-purified both tagged Mic60 versions by Western blot analysis (*Figure 2D*). To rule out the possibility that Mic60 foci represent insoluble aggregates, we solubilized mitochondria isolated from wild type and ΔMICOS cells expressing Mic60-EGFP using the non-ionic detergent, Triton X-100, under relatively low ionic conditions and subjected extracts to ultracentrifugation. As in wild type cells, Mic60-EGFP expressed in ΔMICOS cells remained soluble, indicating that Mic60 foci represent assembled structures (*Figure 2E*). These data indicate that Mic60 self-interacts in the absence of other MICOS members and suggest that Mic60 foci in ΔMICOS may represent self-assemblies.

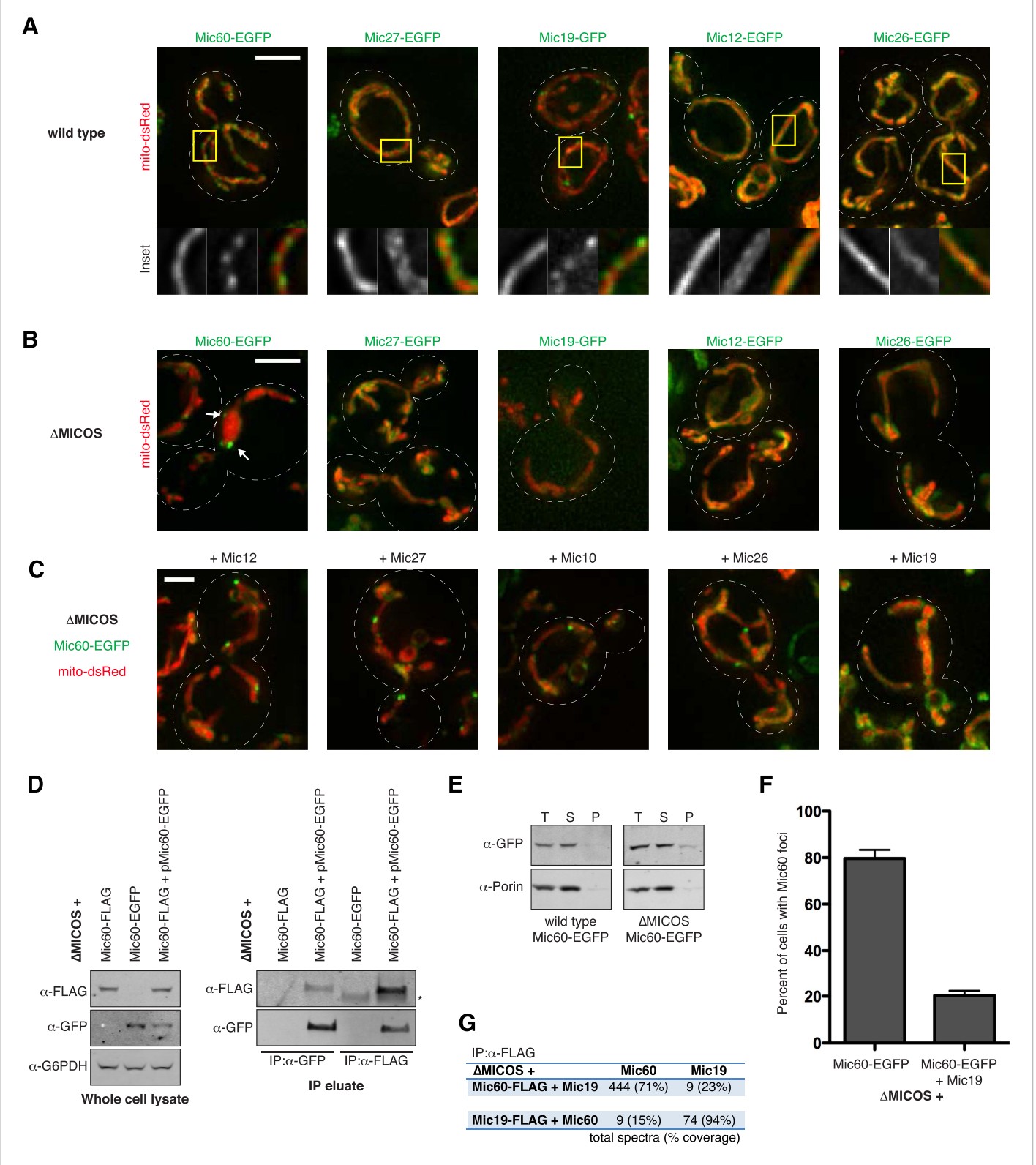

**Figure 2**. Mic60 self-assembles independently and its inner membrane distribution is regulated by Mic19. (**A**) Z-projections of representative deconvolved fluorescence microscopy images are shown from wild type yeast cells expressing the indicated integrated GFP-tagged MICOS protein and mito-DsRed. The yellow box indicates the region of the cell shown in the inset below. The inset displays a single plane. (**B**) Representative images are shown as in (**A**) for ΔMICOS cells expressing the indicated GFP-tagged MICOS proteins re-integrated at their endogenous loci. The arrows mark sites of Mic60 localization to

*Figure 2. Continued*

foci. (**C**) The distribution of Mic60-EGFP in ΔMICOS cells was determined as in (**B**) with the indicated untagged MICOS proteins re-integrated at their endogenous loci. (**D**) Representative Western blot analysis with the indicated antibodies are shown of whole cell lysates (left panel) and immunoprecipitation eluates (IPs; right panel) from ΔMICOS cells expressing either Mic60-FLAG or Mic60-EGFP at the *MIC60* locus, and where indicated, expressing Mic60-EGFP from the *ura3* locus using the *MIC60* promoter (pMic60-EGFP). G6PDH antibody was used as a loading control. IPs were performed with the indicated antibodies. The asterisk marks a band consistent with the size of IgG heavy chain. (**E**) Western blot analysis with the indicated antibodies of total (T), supernatant (S), and insoluble (P) fractions of detergent-solubilized mitochondria isolated from wild type (left) or ΔMICOS (right) cells expressing Mic60-EGFP and centrifuged at 50,000×g for 1 hr, a condition that pellets particles of 60S and greater. (**F**) A graph showing the percentage of cells with detectable Mic60 foci from ΔMICOS cells without and with Mic19 expression as shown in (**B**) and (**C**). Approximately 75 cells from three independent experiments were quantified and data are represented as mean ± SEM. (**G**) Table describing the number of total spectra and protein coverage for the indicated proteins (top) from purifications and mass spectrometry analysis using FLAG antibody from ΔMICOS cell lysate expressing the indicated combinations of Mic60 and Mic19 (left) expressed at their endogenous loci. Data shown are the mean of two independent experiments. Scale bars: (**A**–**B**) 3 μm; (**C**) 2 μm. See also *Figure 2—figure supplement 1*.

The following figure supplement is available for figure 2:

**Figure supplement 1**. Overexpression of Mic60 does not alter its focal localization in ΔMICOS cells.

To test the relationship of Mic60-labeled structures to MICOS, we examined the effect of expression of other MICOS complex subunits re-integrated at their respective loci on Mic60 localization. Expression of Mic10, Mic12, Mic26, or Mic27 had no apparent effect on Mic60 foci formation or frequency (*Figure 2C*). In contrast, expression of Mic19 caused Mic60 to become more distributed within mitochondria as compared to the matrix marker, with only 20% of ΔMICOS + Mic19 cells possessing Mic60-EGFP foci as compared to 80% in cells expressing Mic60-EGFP alone (*Figure 2C,F*). The unique dependence on Mic19 for Mic60 distribution in ΔMICOS cells is consistent with the dependence of Mic19's stability on Mic60 in wild type cells (*Harner et al., 2011*; *Hoppins et al., 2011*; *von der Malsburg et al., 2011*). To gain insight into the molecular basis of the effect of Mic19 on Mic60 distribution, we performed purifications using α-FLAG antibody and mass spectrometry analysis from cross-linked cell extracts of ΔMICOS strains expressing Mic60-FLAG and untagged Mic19, or Mic19-FLAG and untagged Mic60, expressed from their endogenous loci. In each purification, we identified both Mic60 and Mic19 (*Figure 2G*), suggesting that Mic19 interacts directly with Mic60 independently of other MICOS subunits and that Mic19, in part, functions within MICOS to regulate the distribution of Mic60 along the inner membrane.

## Mic10, Mic12, and Mic27 form a second independent MICOS organizing center

Given that Mic60 is able to label discrete structures independently of other MICOS complex members, we tested whether Mic60 was required for the focal distribution of other MICOS subunits. Consistent with the requirement of Mic60 for wild type Mic19 expression, Mic19 in Δ*mic60* cells, as detected using fluorescence microscopy, was relatively faint and uniformly labeled the mitochondrial membrane as compared to the matrix marker mito-DsRed (*Figure 3A* vs *Figure 2A*). In the absence of Mic60, Mic26 also localized uniformly, similar to the matrix marker (*Figure 3A*). In contrast, Mic12 and Mic27 both localized in a non-uniform pattern within mitochondria; this phenotype was most prominent for Mic27, which localized to distinct and relatively bright punctate structures in a majority of cells (*Figure 3A*, marked by arrows, and *Figure 3F*). This suggests that Mic27, and to a lesser extent Mic12, localize to mitochondrial sub-structures in a Mic60-independent manner.

We further characterized the nature of the Mic27 sub-structures by examining the localization of Mic27 in the absence of each of the MICOS subunits in cells. In the case of Δ*mic26 and* Δ*mic12* cells, Mic27 had similar non-uniform mitochondrial labeling as observed in wild type cells (*Figure 3B*). In Δ*mic10* cells, Mic27 uniformly labeled mitochondria and the fluorescence signal was relatively weaker than in wild type cells, consistent with the observation that Mic10 is required for Mic27 stability. However, in addition to Δ*mic60* cells, Mic27 localized to bright foci in Δ*mic19* cells, in which there is stable Mic60 expression (*Harner et al., 2011*) (*Figure 3B*, marked by arrows). This observation is consistent with our data suggesting that Mic60 and Mic19 form a complex and function together in a cooperative manner within MICOS (*Figure 2*). More significantly, because

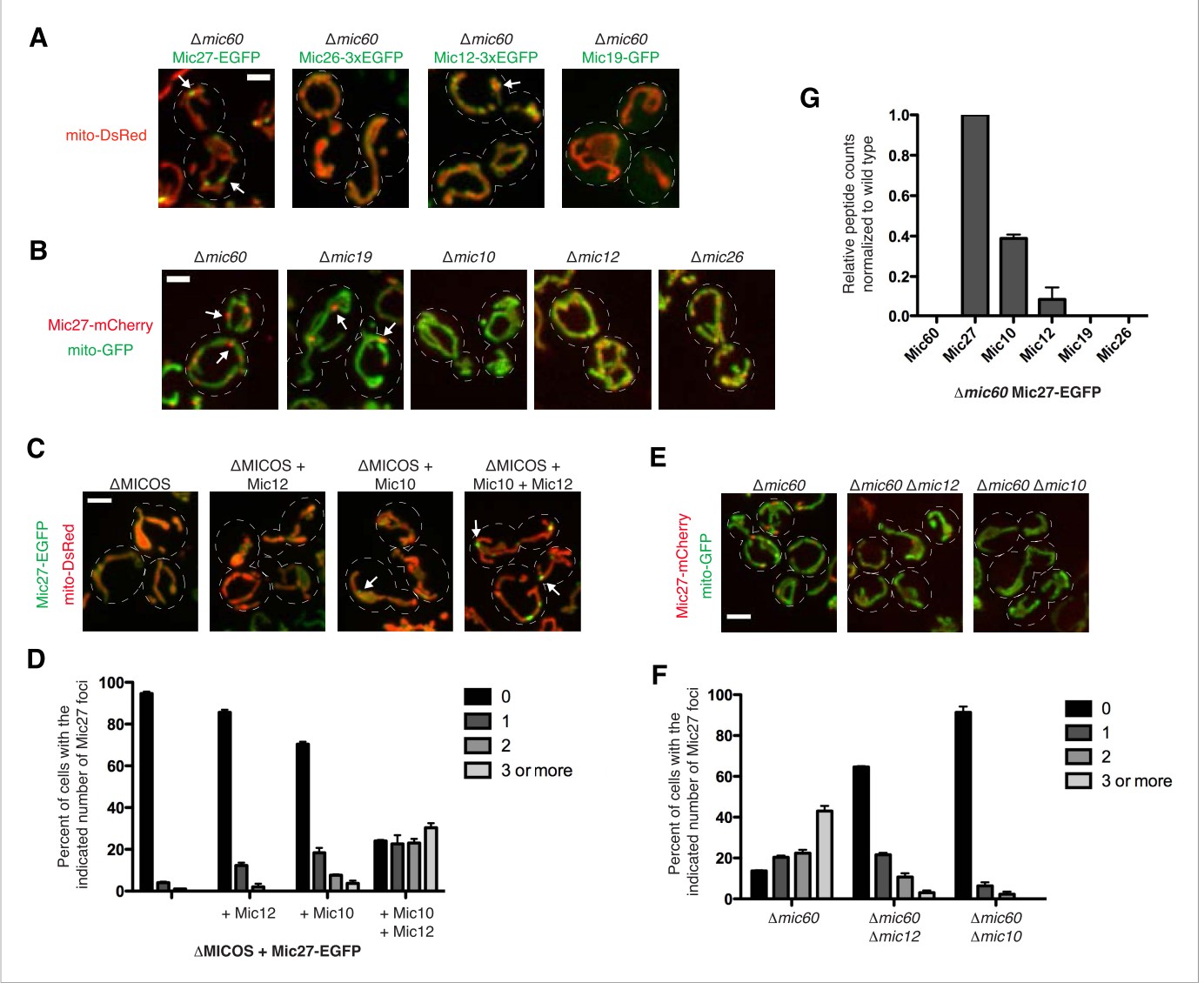

**Figure 3**. Mic10, Mic12, and Mic27 form an independent MICOS subcomplex. (**A**) Confocal fluorescence microscopy z-projections are shown from Δ*mic60* cells expressing the indicated GFP-tagged MICOS proteins expressed at their endogenous loci and the mitochondrial matrix marker mito-DsRed. The arrows shown mark the focal localization of Mic27 and Mic12. (**B**) Localization of Mic27-mCherry compared to the matrix marker mito-GFP was determined in the indicated MICOS deletion cells by confocal fluorescence microscopy as in (**A**). The arrows mark examples of Mic27 foci. (**C**) Images are shown as in (**A**) for ΔMICOS cells expressing Mic27-EGFP and the indicated untagged MICOS proteins expressed at their endogenous loci. The arrows mark Mic27 foci. (**D**) A graph displaying quantification of the percent of cells as in (**C**) with the indicated number of Mic27-EGFP foci per cell. Approximately 75 cells from three independent experiments were counted and data shown are represented as the mean ± SEM. (**E**) Images are shown as in (**B**) for the indicated yeast cells. (**F**) Quantification as in (**D**) for the cells shown as in (**E**). (**G**) A graph depicting the number of total spectral counts identified by mass spectrometry of the indicated MICOS proteins from lysates of Δ*mic60* cells relative to wild type cells from purifications of Mic27-EGFP with GFP antibody. Data shown are the mean and range of two independent experiments. Scale bars: (**A**, **B**) 2 μm; (**C**, **D**) 3 μm. See also *Figure 3—source data 1*.

The following source data is available for figure 3:

**Source data 1**. Table listing spectral count and percent coverage data from mass spectrometry analysis of indicated MICOS protein purifications from the indicated strains.

Mic60 is required for Mic19 stability, it seems likely that the altered localization of Mic27 to more pronounced mitochondrial foci/substructures is a direct consequence of a loss of Mic19 and suggests that these structures represent a second MICOS organizing center.

The formation of the Mic27 substructures requires other MICOS subunits as these structures were not observed when Mic27 is expressed alone in ΔMICOS cells (*Figure 2B*). To determine the minimal MICOS proteins required for Mic27 substructure formation, we re-integrated unmarked MICOS subunits into ΔMICOS cells expressing Mic27-EGFP at its endogenous locus. Consistent with the requirement of Mic10 for Mic27 stability in wild type cells, the addition of Mic10 was sufficient for Mic27 substructures to form, however, their frequency was significantly less than that observed in either Δ*mic19* or Δ*mic60* cells (*Figure 3C,D*). Expression of Mic12 in addition to Mic10, however, supported the formation of Mic27-EGFP substructures at a frequency almost equal to that observed in Δ*mic19* or Δ*mic60* cells (compare *Figure 3D,3F*). Thus, MICOS subunits Mic10 and Mic12 are sufficient for Mic27 substructures to form, suggesting that these three proteins exist as an independent MICOS subcomplex.

To further test this idea, we asked whether Mic10 or Mic12 were required for Mic27 substructure formation in Δ*mic60* cells. Mic27 substructures were largely undetectable in Δ*mic60* Δ*mic10* cells, consistent with the requirement of Mic10 for Mic27 stability (*Figure 3E–F*). We also observed that Mic12 plays a role in Mic27 substructure formation, as the percentage of cells containing Mic27 foci was significantly lower in the absence of Mic12 (*Figure 3F*). Thus, together our data indicate that Mic10 and Mic12 are both necessary and sufficient for Mic27 substructure assembly and suggest that these proteins form an independent MICOS subcomplex.

To directly test this possibility, we purified EGFP-tagged Mic27 from cross-linked lysates of wild type and Δ*mic60* cells using α-GFP antibody and identified MICOS subunits by mass spectrometry-based proteomic analysis. Consistent with previous results, proteomic analysis of purifications of Mic27-EGFP in wild type cells robustly identified the entire MICOS complex as assessed by both the number of peptide spectral counts and percent coverage of MICOS subunits (*Hoppins et al., 2011*) (*Figure 3—source data 1*). To examine Mic27 substructure composition, we directly compared spectral counts of MICOS subunits by normalizing the total number of Mic27 peptide spectral counts purified from Δ*mic60* extracts relative to wild type extracts. As expected, purification of Mic27-EGFP from Δ*mic60* extracts did not identify Mic60 or Mic19; however, significant relative peptide spectral counts were observed for Mic10, and to a lesser extent, for Mic12 (*Figure 3G* and *Figure 3—source data 1*). In total, the data indicate that there are two distinct MICOS organizing centers: Mic60, which may have an intrinsic self-organizational capacity and interacts with Mic19, and a complex containing Mic27, which is dependent on Mic12 and Mic10 for assembly.

## Mic27 assemblies mark cristae junctions

Because of the severe respiratory defect of the ΔMICOS strain and the interrelationship between respiratory complexes and MICOS for generating mitochondrial inner membrane shape (*Hoppins et al., 2011*), we tested whether respiratory complexes are required for Mic60 and Mic27 assembly formation. To block the assembly of mitochondrial respiratory complexes, we generated strains that lack mtDNA (rho[0]) and examined the distribution of the two MICOS subcomplexes as marked by either Mic60 foci (in ΔMICOS + Mic60-EGFP cells) or Mic27 foci (in Δ*mic60* + Mic27-mCherry cells). We observed that Mic60 foci in ΔMICOS persisted in a majority of rho[0] cells (foci in 80% of rho[+] cells vs 64% of rho[0] cells; *Figure 4A–B*). In contrast, although steady state expression level of Mic27 in Δ*mic60* rho[0] cells was similar to in rho[+] cells, Mic27 assemblies were absent in rho[0] cells (foci in 86% of rho[+] cells vs 3% of rho[0] cells; *Figure 4A–B* and *Figure 4—figure supplement 1*). In Δ*mic60* rho[0] cells, the absence of Mic27 foci corresponded to a loss of its interaction with Mic10 and Mic12 as assessed by mass spectrometry-based proteomic analysis of Mic27 purifications (*Figure 4C* and *Figure 4—source data 1A*). The loss of Mic27 subcomplex interactions is due specifically to the combination of mtDNA loss and the absence of Mic60, as Mic27 co-purified with the entire MICOS complex in wild type rho[0] cells as assessed by mass spectrometry analysis of Mic27 purifications (*Figure 4—source data 1B*). These data reveal that the Mic27/Mic10/Mic12 subcomplex is selectively dependent on the presence of mtDNA, suggesting a critical role of respiratory complexes in the assembly and/or stabilization of this MICOS subcomplex.

To test this, we examined the requirement of individual respiratory complexes for Mic27/Mic10/Mic12 subcomplex formation/stability by generating cells deficient for respiratory complex assembly factors for Complex III, IV, or ATP synthase (Δ*cbs1*, Δ*mss51*, or Δ*atp10*, respectively) in Δ*mic60* cells expressing Mic27-EGFP (*Graef and Nunnari, 2011*). Deletion of each respiratory complex assembly

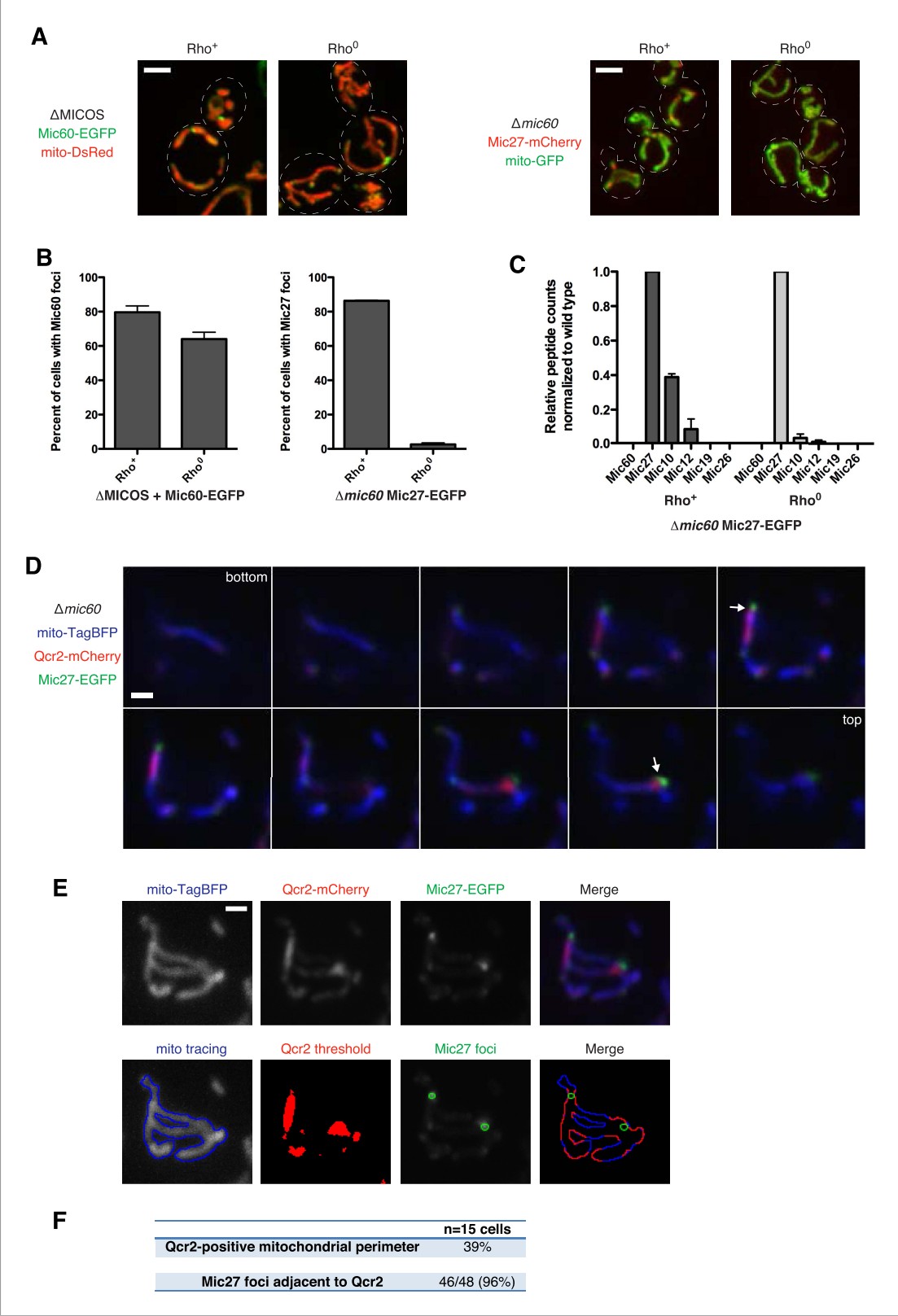

**Figure 4**. Mic27 assemblies mark cristae junctions. (**A**) Z-projections of confocal fluorescence microscopy images of (left) ΔMICOS cells expressing Mic60-EGFP at the *MIC60* locus and mito-DsRed or (right) Δ*mic60* cells expressing Mic27-mCherry at its endogenous locus and mito-GFP. Cells are rho+ or rho0 where indicated. (**B**) Quantification of cells from (**A**) is shown indicating the number of cells with detectable Mic60 or Mic27 foci, as indicated.
*Figure 4. continued on next page*

*Figure 4. Continued*

Approximately 75 cells from three independent experiments were counted and the data shown are represented as the mean ± SEM. Data from rho$^+$ cells for Mic60 and Mic27 are redisplayed from *Figures 2F*, *3F*, respectively. (**C**) A graph depicting the number of total spectral counts identified by mass spectrometry of the indicated MICOS proteins from lysates of Δ*mic60* rho$^+$ and rho$^0$ cells relative to wild type cells from purifications of Mic27-EGFP with GFP antibody. Data shown are the mean and range of two independent experiments and data for rho$^+$ cells are redisplayed from *Figure 3G*. (**D**) Single plane confocal fluorescence microscopy images in 0.4 μm steps through a Δ*mic60* cell expressing Mic27-EGFP and the Complex III marker Qcr2-mCherry at their endogenous loci and the mitochondrial matrix marker mito-TagBFP. Arrows indicate Mic27 foci identified and analyzed in (**E**). (**E**) Top row: individual channels and a merged image of a maximum z-projection image of the cell shown in (**D**). Bottom row: from left to right, a tracing of mitochondria, Qcr2 signal above the threshold detected by the 'Moments' algorithm of ImageJ, Mic27 foci identified in the z-projection, and a merged image indicating regions of the mitochondrial perimeter positive for Qcr2 signal and their position relative to Mic27 foci. (**F**) Quantification of the total percentage of mitochondrial perimeter considered positive for Qcr2 signal and the number of Mic27 foci localized to mitochondrial subregions positive for Qcr2 from cells in (**E**) and *Figure 4—figure supplement 4*. Scale bars: (**A**) 3 μm; (**D–E**) 1 μm. See also *Figure 4—figure supplements 1–5* and *Figure 4—source data 1*.

The following source data and figure supplements are available for figure 4:

**Source data 1**. Tables listing spectral count and percent coverage data from mass spectrometry analysis of indicated MICOS protein purifications from the indicated strains.

**Figure supplement 1**. Steady-state Mic27 expression levels are maintained in Δ*mic60* rho$^0$ cells compared to rho$^+$ cells.

**Figure supplement 2**. Individual respiratory complexes are not essential for Mic27 foci formation.

**Figure supplement 3**. Qcr2 and Atp2 are functional markers of Complex III and V, respectively, and localize to sub-mitochondrial regions in Δ*mic60* cells.

**Figure supplement 4**. Mic27 assemblies localize adjacent to the cristae marker Qcr2.

**Figure supplement 5**. Mic27 assemblies localize adjacent to the cristae marker Atp2.

factor caused a significant decrease in the percentage of cells containing Mic27 foci (from 86% to 44–63%; *Figure 4—figure supplement 2*). This observation indicates that individual respiratory complexes contribute to Mic27/Mic10/Mic12 subcomplex formation/stability, but are not essential. Together, our data indicate that respiratory complexes function in a somewhat redundant manner in the assembly/stability of Mic27/Mic10/Mic12.

The complete absence of respiratory complexes in rho$^0$ cells corresponds to a loss of detectable characteristic cristae as assessed by EM analysis (*Hoppins et al., 2011*). Thus, we considered the possibility that the Mic27/Mic10/Mic12 subcomplex marks cristae junctions. Previously, we observed that cells harboring individual MICOS deletions have a 3–10-fold reduction in the frequency of cristae junctions, but that junction architecture is not altered (*Hoppins et al., 2011*). Thus, the relatively discrete nature and low copy number of Mic27 assemblies in Δ*mic60* cells relative to the abundant assemblies in wild type cells is also consistent with its localization at cristae junctions. A direct assessment of whether Mic27-EGFP localizes to cristae junctions in Δ*mic60* cells by thin section immuno-EM was not successful due to technical challenges posed by the low copy number of cristae junctions. As an alternative, we utilized fluorescent protein fusions of respiratory complex constituents that selectively localize to cristae (*Wurm and Jakobs, 2006*). In wild type cells, functional markers for Complex III (Qcr2-mCherry) and ATP synthase (Atp2-mCherry) were relatively uniformly distributed as compared to the matrix marker, consistent with previous observations (*Wurm and Jakobs, 2006*) (*Figure 4—figure supplement 3A–3B*). In contrast, in Δ*mic60* cells, both Qcr2-mCherry and Atp2-mCherry possessed a non-uniform distribution and concentrated in discrete subregions of mitochondria as compared to the matrix marker (*Figure 4—figure supplement 3C*). In addition, Atp2-EGFP fluorescence signal marked edges of

Qcr2-mCherry-labeled subregions of Δ*mic60* cells (*Figure 4—figure supplement 3D–3E*). EM analysis of ATP synthase localization in different cell types indicates that it selectively localizes in multimeric ribbons to the curved edges of cristae (*Strauss et al., 2008*; *Acehan et al., 2011*). Thus, the localization patterns of Atp2 in subregions marked by Qcr2 strongly suggests that these are lamellar, stacked cristae as visualized by EM in Δ*mic60* cells.

We tested whether Mic27 structures were localized adjacent to cristae by examining whether Mic27 foci were spatially linked to Qcr2-labeled subregions in Δ*mic60* cells. Significantly, we observed that a majority of Mic27 foci were adjacent to regions labeled by Qcr2 in Δ*mic60* cells (96% of Mic27 foci were localized close to Qcr2, which marked less than 40% of the mitochondrial perimeter, n = 15 cells; *Figure 4D–F* and *Figure 4—figure supplement 4*). Mic27 foci also were spatially linked to Atp2 labeled subregions in Δ*mic60* cells (*Figure 4—figure supplement 5*). Taken together our data strongly suggest that independently of Mic60, at the boundary membrane, the Mic27/Mic10/Mic12 subcomplex is positioned at cristae junctions.

## Cardiolipin is required for Mic27 assemblies

We examined the role of the mitochondrial lipid environment on MICOS. In this context, the paralogous Mic27 and Mic26 contain an Apolipoprotein O-like lipid-binding domain, and the mammalian homolog, MIC27/APOOL, has been shown to specifically bind to cardiolipin in vitro (*Weber et al., 2013*). In addition, members of the MICOS complex have a strong negative genetic interaction with members of the cardiolipin synthesis pathway, further suggesting that MICOS and cardiolipin act together to maintain mitochondrial function (*Hoppins et al., 2011*). We therefore asked whether cardiolipin affected the assembly/stability of the Mic27 subcomplex in mitochondria. We examined both the localization of Mic27 and mitochondrial morphology in wild type cells and cells lacking the mitochondrial cardiolipin synthesis enzyme, Crd1. Deletion of *CRD1* led to minor mitochondrial morphology and respiratory growth defects in cells, consistent with published observations (*Jiang et al., 2000*; *Chen et al., 2010*) (*Figure 5A,D,E*). In Δ*crd1* cells, the non-uniform localization pattern of Mic27 was similar to wild type cells and consistently, mass spectrometry analysis of purifications of Mic27 from Δ*crd1* cell extracts indicated that the MICOS complex was intact (*Figure 5A,C*, and *Figure 5—source data 1*). In contrast, we found that in Δ*mic60* Δ*crd1* cells, Mic27-EGFP foci were largely absent and Mic27 stability was reduced as compared to levels in wild type, Δ*mic60*, or Δ*crd1* cells (foci in 87% of Δ*mic60* cells vs 2% of Δ*mic60* Δ*crd1* cells; *Figure 5A–B* and *Figure 5—figure supplement 1A*). The requirement of cardiolipin for Mic27 stability and Mic27 assemblies was specific, as Mic60 foci formation was not significantly affected in ΔMICOS Δ*crd1* cells as compared to ΔMICOS cells (*Figure 5—figure supplement 1B*). Consistent with loss of Mic27 assemblies, purification and mass spectrometry analysis of Mic27 from cross-linked Δ*mic60* Δ*crd1* cell extracts indicates a loss of interaction of Mic27 with the MICOS complex (*Figure 5C* and *Figure 5—source data 1*). Together, these data indicate that the association of Mic27 to the Mic27/Mic10/Mic12 subcomplex is dependent on cardiolipin and suggest that cardiolipin is required for its localization at cristae junctions.

Because Δ*mic60* Δ*crd1* cells are deficient for both Mic60 and Mic27 assemblies, we asked whether their growth and mitochondrial morphological characteristics are similar to ΔMICOS cells. We observed that Δ*mic60* Δ*crd1* and ΔMICOS have comparable growth defects when utilizing glycerol as a non-fermentable carbon source (*Figure 5D*). In contrast, Δ*mic27* Δ*crd1* cells did not have a synthetic growth defect on glycerol media (*Figure 5—figure supplement 1C*). Additionally, deletion of *CRD1* in ΔMICOS cells did not further compromise growth on non-fermentable carbon sources (*Figure 5D*, compare growth of ΔMICOS Δ*crd1* vs ΔMICOS strains). Consistently, mitochondrial morphology of Δ*mic60* Δ*crd1* cells was more severely affected than that observed in Δ*mic60* cells and had similar characteristics to those observed in ΔMICOS cells, with a substantial increase in the percent of cells with bulbous morphology (9% of Δ*mic60* cells compared to 36% Δ*mic60* Δ*crd1* cells) and a corresponding decrease in the percent of cells with tubular and lamellar morphology (*Figure 5A,E*). Together, these data indicate that in cells the absence of both Mic60 and cardiolipin partially phenocopies deletion of the entire MICOS complex and suggest that cardiolipin, in addition to its role in the assembly of respiratory chain supercomplexes, acts to promote the assembly of Mic27 into the Mic27/Mic10/Mic12 subcomplex and its localization to cristae junctions. Importantly, the genetic interaction between *CRD1* and *MIC10* differs from that of *CRD1* and *MIC27*, suggesting that there are complex interactions

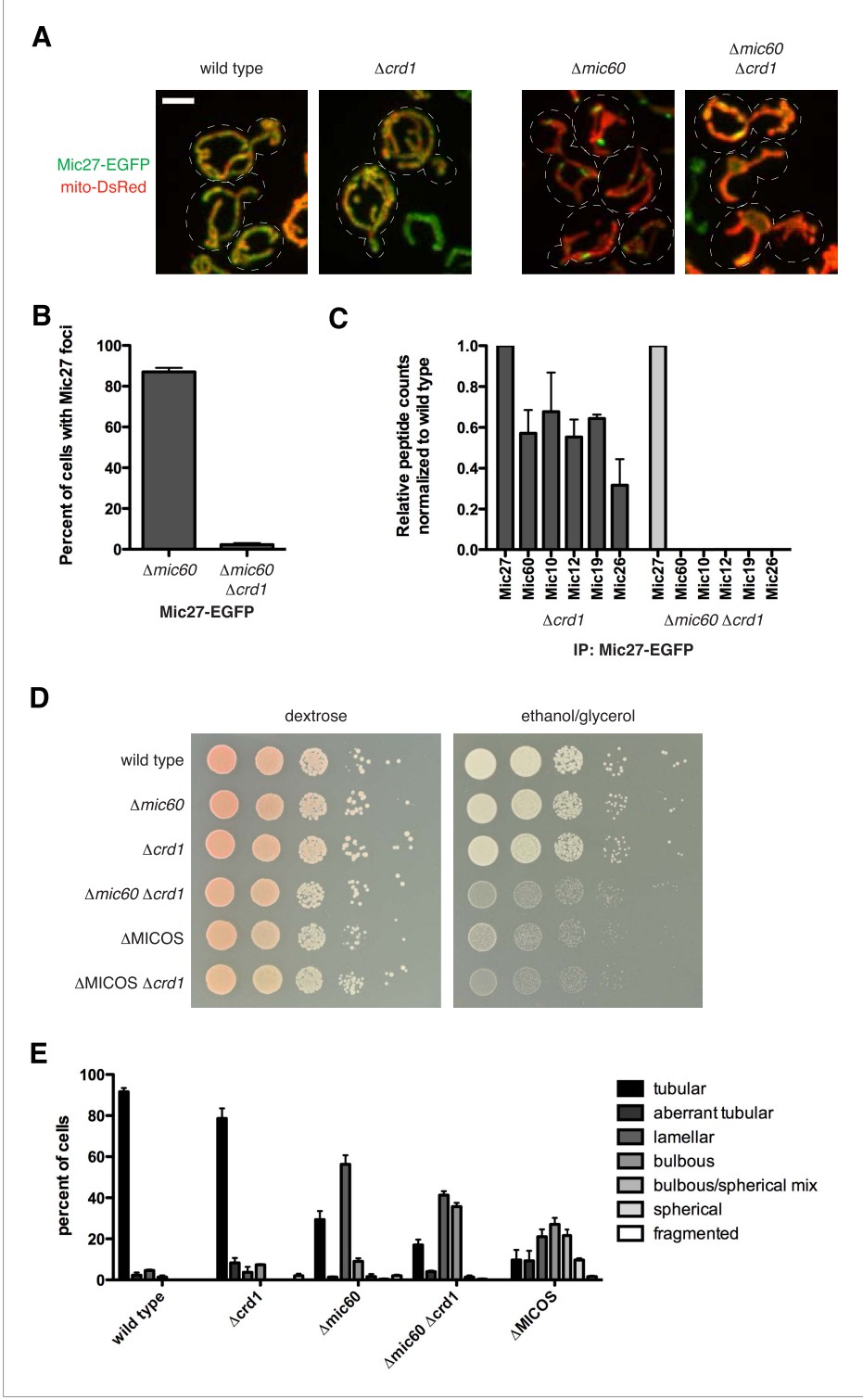

**Figure 5**. Cardiolipin is required for stability of the Mic10/Mic12/Mic27 subcomplex. (**A**) Images are shown of confocal fluorescence microscopy z-projections of the indicated cells expressing Mic27-EGFP from its endogenous locus and mito-DsRed. (**B**) Quantification of the percent of cells with detectable Mic27 foci in the indicated cells. Data shown are represented as the mean ± SEM from three independent experiments with approximately 75 cells counted per experiment. (**C**) A graph depicting the number of total spectral counts identified by mass spectrometry of the indicated MICOS proteins from lysates of Δ*crd1* and Δ*mic60* Δ*crd1* cells relative to wild type cells from purifications of Mic27-EGFP with GFP antibody. (**D**) Serial dilutions of the indicated yeast cells plated on media
*Figure 5. continued on next page*

*Figure 5. Continued*
containing glucose (left) and the non-fermentable carbon source, glyercol (right). (**E**) Quantification of mitochondrial morphologies from the indicated cells expressing mito-DsRed were categorized. Approximately 75–100 cells from three independent experiments were quantified and data are represented as mean ± SEM. Data for wild type and ΔMICOS cells are redisplayed from *Figure 1C*. Scale bars: 3 μm. See also *Figure 5—figure supplement 1* and *Figure 5—source data 1*.

The following source data and figure supplement are available for figure 5:

**Source data 1**. Table listing spectral count and percent coverage data from mass spectrometry analysis of indicated MICOS protein purifications from the indicated strains.

**Figure supplement 1**. Interactions between cardiolipin synthesis and MICOS.

both within and between each MICOS subcomplex and their mitochondrial environment (*Figure 5—figure supplement 1C*).

## Respiratory complex III and IV contribute to ΔMICOS phenotypes

To understand the role of MICOS in mitochondrial respiratory function, we explored the basis of the ΔMICOS respiratory growth defect. We isolated purified mitochondria from wild type, Δ*mic60*, and ΔMICOS strains, and assessed mitochondrial respiratory complex function in vitro. Mitochondrial respiratory complexes and supercomplexes remained largely intact in both Δ*mic60* and ΔMICOS mitochondria as assessed by Blue Native PAGE (BN-PAGE) analysis relative to mitochondria isolated from wild type cells (*Figure 6A*). Likewise, both ATP synthase dimerization and activity were unaffected in either single MICOS deletions or ΔMICOS as assessed by an in gel activity assay (*Figure 6C*). However, despite the presence of mitochondrial respiratory complexes, we detected deficiencies in cytochrome-*c* oxidase (Complex IV) activity specifically in ΔMICOS and not in Δ*mic60* (*Figure 6B*). Using two-dimensional gel analysis, we determined that the composition of respiratory complexes in Δ*mic60* and ΔMICOS cells was indistinguishable from wild type cells (*Figure 6D*). These data indicate that in the absence of the MICOS complex, respiratory complexes can assemble, but do not function optimally. It is possible that in the absence of the MICOS complex, the respiratory defect results from a severe decrease in cristae junctions and consequent disorganization of the inner membrane structure. Alternatively, but not exclusively, MICOS could play a more direct role in the organization of respiratory complexes in the inner membrane.

To further test this idea, we examined the relationship between respiratory complexes and the mitochondrial morphology defect in ΔMICOS cells. Specifically, we asked whether mitochondrial morphology was altered in ΔMICOS rho$^0$ cells, which lack mitochondrial respiratory complexes. Strikingly, ΔMICOS rho$^0$ cells exhibited relatively normal, tubular mitochondrial morphology as assessed by fluorescence microscopy of matrix-targeted dsRed as compared to ΔMICOS rho$^+$ cells (10% rho$^+$ cells vs 95% rho$^0$ cells exhibited tubular mitochondrial morphology; *Figure 6E,G*). As expected, by electron microscopy, the inner membrane of ΔMICOS rho$^0$ cells lacked detectable cristae and were frequently septated, suggesting that mitochondrial inner membrane fusion is compromised likely as a secondary consequence of reduced respiration (*Figure 6F*) (*Sauvanet et al., 2012*). Thus, in the absence of the MICOS complex, expression of respiratory complexes causes inner membrane disorganization, and consequently, severe and diverse abnormal mitochondrial morphologies. Consistently, loss of ATP synthase was previously shown to suppress the predominant lamellar mitochondrial morphology defect of Δ*mic60* cells (*Hoppins et al., 2011*). Therefore, we asked whether deletion of individual respiratory complexes alleviated the ΔMICOS mitochondria morphological defect by deleting genes for essential assembly factors of Complex III (Δ*cbs1*), Complex IV (Δ*mss51*), and ATP synthase (Δ*atp10*) from ΔMICOS. As in Δ*mic60* cells, deletion of ATP synthase assembly selectively suppressed the abnormal mitochondrial lamellar phenotype of ΔMICOS, but the bulbous or spherical morphologies were still present in the cell population. Only 25% of ΔMICOS Δ*atp10* cells had normal tubular morphology, compared to 10% in ΔMICOS alone, suggesting distinct morphological phenotypes present in ΔMICOS are caused by separate factors (*Figure 6E,G*). Consistent with this, deletion of either Complex III or IV assembly

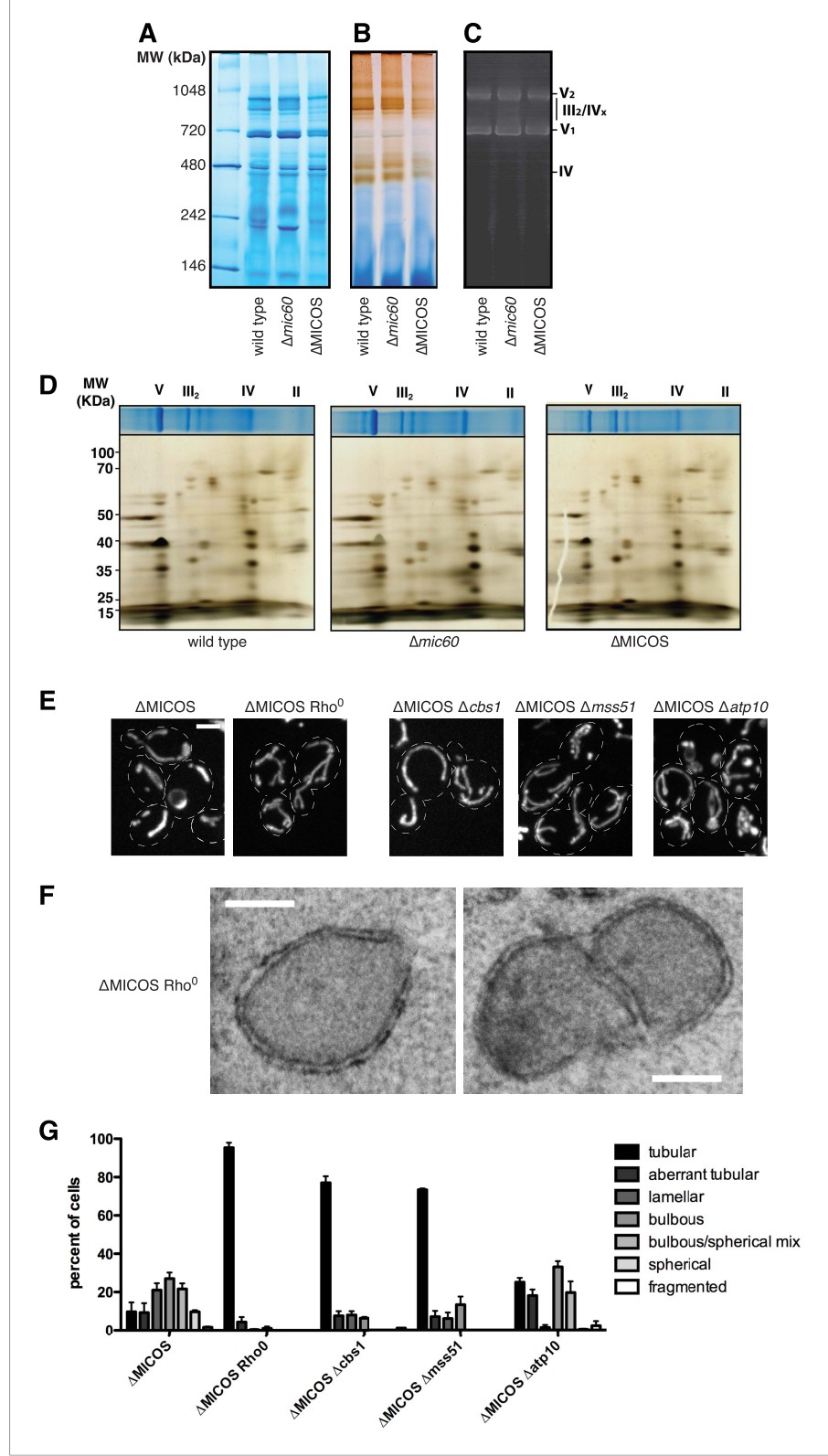

**Figure 6**. Complex III and IV selectively contribute to the ΔMICOS phenotypes. (**A**) A Coomassie-stained Blue Native-PAGE gel of digitonin-solubilized mitochondria isolated from the indicated strains. Labels indicate approximate sizes of the indicated respiratory complexes and supercomplexes. (**B–C**) In-gel activity assays for samples run as in (**A**) for Complex IV activity (**B**) and ATP synthase activity (**C**). Molecular weight markers are shown

*Figure 6. Continued*
on the left. (**D**) Top: Coomassie-stained Blue Native PAGE (BN-PAGE) gels of DDM-solubilized mitochondria isolated from the indicated strains. Labels indicate approximate sizes of the indicated respiratory complexes. Bottom: BN-PAGE analysis was followed by SDS-PAGE in a second dimension and silver staining analysis. Molecular weights markers are shown on the left. (**E**) Z-projections of confocal fluorescence microscopy images of the indicated yeast cells expressing mito-DsRed are shown. (**F**) Representative electron microscopy images are shown of chemically fixed ΔMICOS rho⁰ cells. (**G**) A graph displaying categorization of mitochondrial morphologies from the indicated cells imaged as in (**E**). Approximately 75–100 cells from three independent experiments were quantified and data are represented as mean ± SEM. Data for ΔMICOS cells are redisplayed from *Figure 1C*. Scale bars: (**E**) 3 µm; (**F**) 200 nm.

factors more significantly alleviated all types of ΔMICOS morphological defects, as ~75% of ΔMICOS Δ*cbs1* or ΔMICOS Δ*mss51* cells had normal tubular mitochondrial morphology (*Figure 6E,G*). These observations, together with the reduced Complex IV activity in ΔMICOS cells, suggest that assembled respiratory Complex III and IV are not properly organized and/or positioned in ΔMICOS, leading to morphological and respiratory growth defects.

## Mic19 mediates the interaction between the Mic60 and Mic27 assemblies at cristae junctions

Our data indicate that, although there is functional redundancy within the MICOS complex, there are functional interrelationships between the Mic60 and Mic27/Mic10/Mic12 MICOS subcomplexes, respiratory complexes, and cardiolipin that are required for proper inner membrane organization, mitochondrial morphology, and respiratory function. To address these interrelationships, we examined how the two MICOS subcomplexes connect and the role of this connection in the proper maintenance of cristae junction number and organization in wild type cells.

We considered Mic19 as it has a functional relationship to each MICOS subcomplex; Mic19 disperses Mic60 foci in ΔMICOS and the absence of Mic19 specifically results in the formation of Mic27 focal assemblies in cells (*Figures 2, 3*). These data are consistent with a possible role of Mic19 as an assembly regulator of both MICOS subcomplexes. To test whether Mic19 functions as a connector, we purified tagged Mic60 and Mic27 from cross-linked Δ*mic19* extracts and identified interacting partners using mass spectrometry analysis, normalizing the total number of Mic60 or Mic27 peptide spectral counts purified from Δ*mic19* extracts relative to wild type extracts. Mass spectrometry analysis of Mic60 purifications indicated that relative to wild type extracts, the absence of Mic19 results in a significant loss of Mic60 interactions with other MICOS components (*Figure 7A* and *Figure 7—source data 1*). By comparison, mass spectrometry analysis of Mic27 purifications from Δ*mic19* extracts indicated that Mic27 maintains its interaction with Mic10 relative to wild type extracts (*Figure 7A* and *Figure 7—source data 1*). Together, our data support a model in which the soluble intermembrane space protein Mic19 functions as a primary connector between the two distinct MICOS organizing centers: Mic60 and Mic27/Mic10/Mic12.

We next addressed the hierarchical relationship between the Mic60 and Mic27/Mic10/Mic12 MICOS subcomplexes. Mic60 assembles independently of the MICOS complex and its interactions with other MICOS members are significantly reduced in Δ*mic19* cells (*Figures 2B, 7A*). Consistent with these data, we observed that Mic60 localizes to focal assemblies in Δ*mic19* cells (*Figure 7B*). We tested the relationship between Mic60 and Mic27/Mic12/Mic10 assemblies in Δ*mic19* cells expressing Mic60-EGFP and Mic27-mCherry. We found that the focal localization of Mic60 and Mic27 overlapped, suggesting Mic60 assemblies also mark cristae junctions (*Figure 7C*, see arrows). Consistently, Mic60 assemblies in Δ*mic19* cells localized adjacent to discrete mitochondrial subregions labeled by the cristae markers Qcr2 and Atp2, further supporting their localization to cristae junctions (*Figure 7D–E* and *Figure 7—figure supplement 2*). Mic60 assemblies in Δ*mic19* cells persist in the absence of mtDNA and cardiolipin synthesis (*Figure 7B* and *Figure 7—figure supplement 1*). Thus, while it is possible that the co-localization of Mic60 with Mic27 in Δ*mic19* cells is due to direct interactions between the Mic60 and Mic27/Mic10/Mic12 subcomplexes, the behavior of Mic60 assemblies in Δ*mic19* cells is consistent with that of Mic60 assemblies in ΔMICOS cells. These observations suggest that Mic60 structures may independently mark cristae junctions in cells.

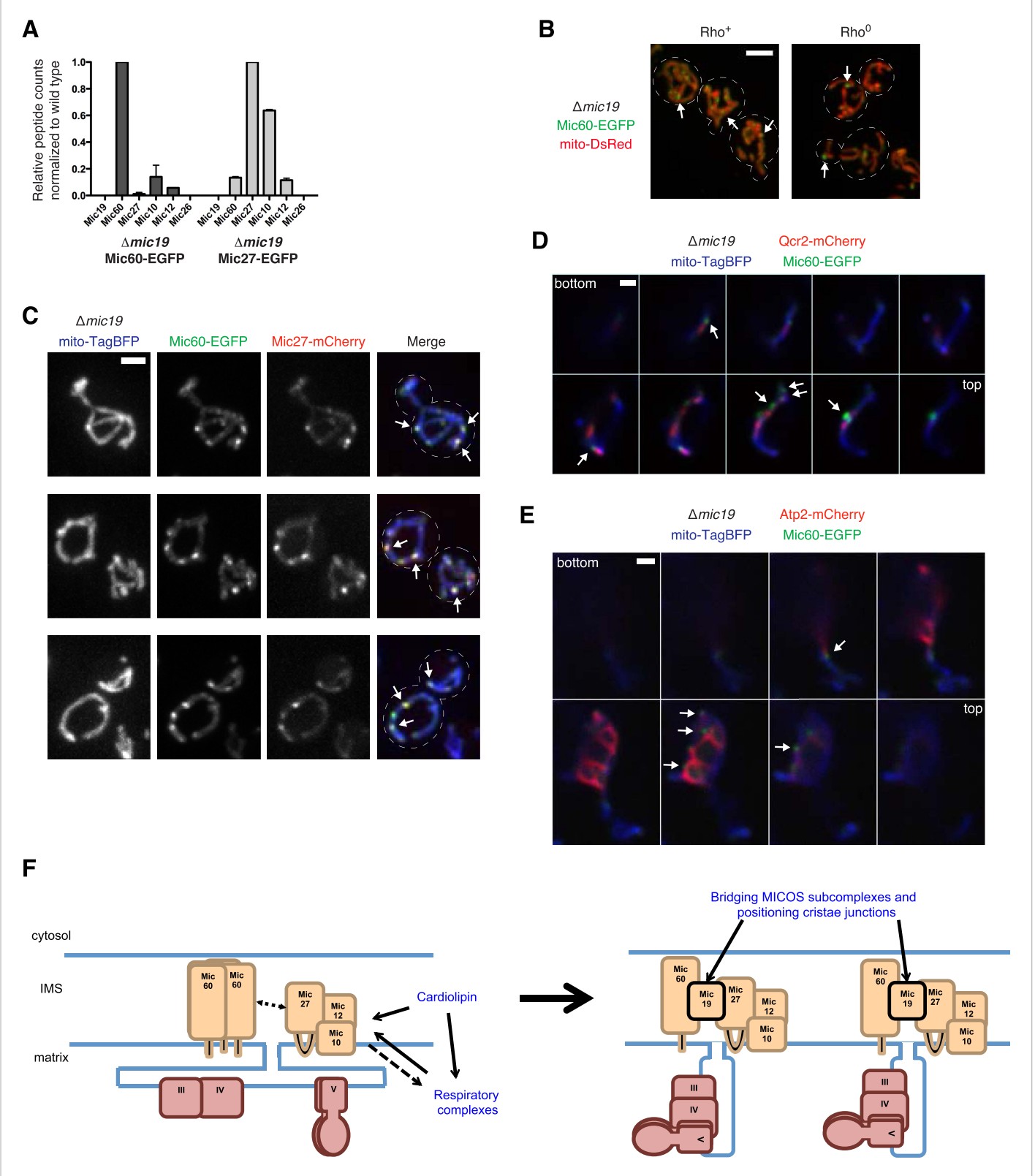

**Figure 7**. Mic19 mediates an interaction between the Mic60 and Mic10/Mic12/ Mic27 subcomplexes. (**A**) A graph depicting the number of total spectral counts identified by mass spectrometry of the indicated MICOS proteins from lysates of Δ*mic19* cells relative to wild type cells from purifications of Mic60- or Mic27-EGFP with GFP antibody. Data shown are the mean and range of two independent experiments. (**B**) Images are shown of confocal fluorescence microscopy z-projections of Δ*mic19* rho[+] (left) and rho[0] (right) cells expressing Mic60-EGFP from the *MIC60* locus and mito-DsRed.
*Figure 7. continued on next page*

*Figure 7. Continued*

Arrows mark focal localization of Mic60. (**C**) Z-projections are shown of confocal fluorescence microscopy images of Δ*mic19* cells expressing mito-TagBFP as well as Mic60-EGFP and Mic27-mCherry expressed from their endogenous loci. Arrows indicate sites of Mic60 and Mic27 colocalization. (**D**) Single plane confocal fluorescence microscopy images in 0.4 μm steps through a Δ*mic19* cell expressing Mic60-EGFP and the Complex III marker Qcr2-mCherry at their endogenous loci and the mitochondrial matrix marker mito-TagBFP. Arrows indicate the position of Mic60 foci. (**E**) As in (**D**) for cells expressing Atp2-mCherry. (**F**) A schematic model of the organization and roles of MICOS and its constituent subcomplexes. The model also depicts the interrelationship between MICOS, cardiolipin, and respiratory complexes, and the coordination of these factors in generating mitochondrial inner membrane organization and cristae architecture. Scale bars: (**B**) 3 μm; (**C**) 2 μm; (**D**, **E**) 1 μm. See also *Figure 7—figure supplements 1–2* and *Figure 7—source data 1*.

The following source data and figure supplements are available for figure 7:

**Source data 1**. Table listing spectral count and percent coverage data from mass spectrometry analysis of indicated MICOS proteins.

**Figure supplement 1**. Cardiolipin synthesis is not required for Mic60 foci formation in Δ*mic19* cells.

**Figure supplement 2**. Mic60 assemblies localize adjacent to the cristae markers Qcr2 and Atp2.

Their persistence in ΔMICOS rho⁰ cells, which lack detectable cristae further suggests that Mic60 functions upstream of the Mic27/Mic10/Mic12 subcomplex to position nascent sites of cristae junction formation within mitochondria (*Figures 4A, 6F*). In the context of our data supporting a role of Mic19 as a MICOS subcomplex regulator and connector, we propose that Mic19 coordinately regulates the assembly of MICOS to dictate the copy number and distribution of cristae in the inner membrane.

## Discussion

Our analysis indicates that MICOS consists of two subcomplexes, which localize to cristae junctions (*Figure 7F*). Examination of the assembly pathways for each subcomplex has illuminated their relative roles in cristae biogenesis. The Mic27/Mic12/Mic10 subcomplex is dependent on the inner membrane phospholipid cardiolipin and on the expression of respiratory complexes. The second MICOS subcomplex consists of a Mic60 multimer, which assembles in a manner independent of cardiolipin, respiratory complexes, and other MICOS proteins. Although each subcomplex assembles independently, the cardiolipin- and respiratory complex-independent nature of Mic60 assemblies suggest that there is a hierarchical nature to MICOS assembly, where Mic60 functions upstream of the Mic27/Mic10/Mic12 subcomplex in MICOS assembly. In this context, we observed Mic60 assemblies in cells without mtDNA, which lack detectable cristae by electron microscopic analysis. These observations suggest that Mic60 assemblies direct the position of nascent cristae junction sites.

The underlying molecular basis for how Mic60 assemblies are positioned within mitochondria is not currently understood, but is likely critical for inner membrane organization. One possibility is that outer membrane-associated determinants direct the biogenesis and/or placement of Mic60 assemblies. Consistent with this, Mic60 independently interacts with components of the mitochondrial import and sorting machinery complex (SAM/TOB) through its conserved 'mitofilin' domain (*Korner et al., 2012*; *Zerbes et al., 2012*). These interactions may allow Mic60 to facilitate proper spacing between the inner boundary and outer membranes and/or to organize these specific outer membrane components relative to inner membrane cristae. Mitochondrial internal determinants may also contribute to the assembly and position of Mic60, such as the mtDNA-independent replisome, which could function to coordinate cristae biogenesis with nucleoid placement (*Meeusen and Nunnari, 2003*). Alternatively, but not exclusively, it is possible that Mic60 assembles at mitochondrial lipid sub-domains positioned, for example, by contacts between mitochondria and the ER (*Kornmann et al., 2009*).

The role of the Mic27/Mic12/Mic10 subcomplex is suggested by its dependence on cardiolipin and respiratory complexes. Cardiolipin is also required for the assembly and activity of respiratory complexes and for the formation of respiratory supercomplexes, which are processes that do not depend on MICOS. Thus, our observations suggest that Mic27/Mic12/Mic10 may instead directly sense cardiolipin-rich subdomains of mitochondria to coordinate the non-redundant actions of MICOS and cardiolipin in the respiratory complex-dependent formation of cristae junctions. Consistent with this,

Mic26 and Mic27 both share a conserved Apolipoprotein O-like domain, which in other contexts bind to lipids. We could not detect a role for Mic26 in MICOS subcomplex assembly, although its domain architecture suggests it may also function to coordinate MICOS with the mitochondrial lipid environment. In addition, although we failed to generate a functional Mic10 fluorescent protein to directly examine its localization and behavior, the observation that Mic10 is required for Mic27 stability suggests that it plays a central role in directing the formation and function of the Mic27/Mic12/Mic10 subcomplex.

The dependence of the Mic27/Mic12/Mic10 subcomplex on respiratory complexes indicates that in addition to direct roles in cristae biogenesis, they also intrinsically contribute to this biogenesis pathway via MICOS assembly. The reciprocal is not true as we find that MICOS is not required for respiratory supercomplex assembly. However, in the absence of MICOS, the activity of cytochrome-*c* oxidase Complex IV is somewhat reduced and, more importantly, cells lacking MICOS exhibit a severe respiratory defect. One explanation for this severe defect is that MICOS's role in the regulation of cristae copy number is important for producing an optimal distance within cristae between the electron transporting Complex III/IV supercomplexes, localized at lamellar regions, and ATP synthase oligomers localized to the curved edges for efficient oxidative phosphorylation (*Rieger et al., 2014*). Our data also implicate Complex III and IV directly in the nucleation of cristae and in the process of cristae biogenesis. This is supported by the requirement of Complex III and IV for the generation of aberrant mitochondrial morphology in the absence of MICOS (*Figure 6*). The basis for this observation may lie in the ability of Complex III and IV to form large cardiolipin-dependent supercomplexes, which could act as diffusion traps for the cristae proteome and lipidome (*Wilkens et al., 2013*). Recent work also implicates a role for the inner membrane fusion dynamin, OPA1, in the regulation of mitochondrial inner membrane shape and respiratory supercomplex assembly (*Cogliati et al., 2013*). Thus, future work will be required to understand the mechanistic relationships between MICOS and respiratory complexes and OPA1 in the process of inner membrane organization.

Our analysis provides insight into the underlying mechanism of cristae copy number and distribution control. A typical wild type mitochondrion possesses many cristae positioned and distributed along the mitochondrial boundary inner membrane. We find that the only soluble MICOS protein, Mic19 functions as the key component directing the inner membrane distribution of each MICOS subcomplex. We propose that through this distributive action and its role as a connector/adaptor, Mic19 regulates the copy number of the MICOS complex and, consequently, the copy number and position of cristae junctions. In this context, it is interesting to note that Mic19 has two mammalian homologs (MIC19 and MIC25), with conserved coiled-coil-helix motifs, consistent with a role as a key regulatory component of MICOS and suggesting a structural basis for its role as a connector.

Our data indicate that the primary function of MICOS is to stabilize, position and control the copy number of cristae junctions to organize the inner membrane into an efficient respiratory machine. Aberrant cristae morphology, very similar to what we observe in cells devoid of MICOS, is a defining feature of a diverse array of human diseases and of aging, and cristae reorganization is a central facet of cytochrome *c* release in the mediation of apoptotic cell death. Thus, our observations suggest that MICOS will play a significant role in human pathogenesis.

## Materials and methods

### Strains and plasmids

All yeast strains described were constructed in the W303 genetic background (*ade2–1*; *leu2–3*; *his3–11, 15*; *trp1–1*; *ura3–1*; *can1–100*). All deletions (except those used to generate the ΔMICOS strain, see below) were generated using PCR-based homologous recombination replacing the entire ORF of targeted genes with the kanMX6, HIS3MX6, or NatMX6 cassettes (*Longtine et al., 1998*; *Schuldiner et al., 2006*). All C-terminal tags were generated using PCR-based targeted homologous recombination using the following cassettes, and are referred to throughout the text using the description in parentheses: GFP(S65T)::HIS3MX6 (GFP), yEGFP::Kan (EGFP), yEGFP::SpHIS5 (EGFP), yEGFP::CaURA3 (EGFP), 3x-yEGFP::CaURA3 (3x-EGFP), ymCherry::HIS3MX6 (mCherry), ymCherry::Kan (mCherry), and 3xFLAG::HIS3MX6 (FLAG) (*Longtine et al., 1998*; *Sheff and Thorn, 2004*; *Hoppins et al., 2011*; *Graef et al., 2013*). All yeast transformations were performed by the lithium acetate method and transformants were selected on the appropriate media and verified by PCR, and where appropriate, gene expression. Haploid strains containing multiple tags, deletions, or

combinations thereof were generated either by mating and sporulation, or by sequential PCR-based homologous recombination. Rho$^0$ versions of strains were generated by growth in YPD containing 25 µg/ml ethidium bromide for ~48 hr.

The ΔMICOS strain was generated using the Cre-*lox P* system described previously (*Guldener et al., 1996*). Briefly, the *loxP-kanMX-loxP* cassette was PCR amplified from pUG6, integrated into the desired locus by homologous recombination replacing the ORF of the desired MICOS gene, and verified by PCR. The resulting strain was transformed with pSH47, which expresses the Cre recombinase under the control of a *GAL1* promoter. Cre was expressed by growth in YPGal for ~16 hr. Isolated colonies were tested for loss of G418 resistance and further verified by PCR. Finally, loss of the pSH47 plasmid was selected for by growth on 5-fluoroorotic acid. This process was performed sequentially for each MICOS ORF in the following order: *MIC10, MIC60, MIC19, MIC27, MIC26,* and *MIC12*.

Untagged MICOS proteins were reintroduced into ΔMICOS in two steps. First, the kanMX6, TRP1, or HIS3MX6 selection cassettes were introduced after the stop codon of a MICOS ORF in a wild type strain using PCR-based targeted homologous recombination. Then, genomic DNA was prepared from these strains, and the entire cassette containing the MICOS ORF, selection cassette, and regions upstream and downstream of the ORF was PCR-amplified and reintroduced into the ΔMICOS strain. In the case of reintroduction of tagged MICOS proteins, the same procedure was performed using genomic DNA isolated from the tagged strains described above.

pYX142-DsRed (referred to as 'mito-DsRed') and pYX142-mtGFP (referred to as 'mito-GFP') were previously described (*Westermann and Neupert, 2000*; *Friedman et al., 2011*). To construct pVT100u-mito-TagBFP (referred to as 'mito-TagBFP'), yeast codon optimized TagBFP was synthesized de novo as a gBlock (IDT), digested and inserted into the KpnI/XhoI sites of pVT100u-mtGFP (*Westermann and Neupert, 2000*), replacing GFP. All mitochondria plasmid-containing yeast strains were generated by lithium acetate transformation and selection on synthetic dextrose (SD) media.

To construct pRS306-Mic60-EGFP (referred to as 'pMic60-EGFP'), a DNA fragment containing the *MIC60* promoter, the Mic60 ORF fused to EGFP, and the *ADH* terminator was PCR amplified from genomic DNA isolated from a yeast strain expressing Mic60-EGFP at the *MIC60* locus. This fragment was digested and cloned into the BamHI/NotI sites of pRS306 (*Sikorski and Hieter, 1989*). To construct pRS305-Rim1-GFP (referred to as 'Rim1-GFP'), a GFP tag was first integrated at the C-terminus of the Rim1 ORF using pFA6a-GFP(S65T)-kanMX6. A cassette containing the *RIM1* promoter, the Rim1 ORF (including its intron), GFP(S65T), and the *ADH* terminator was PCR amplified from genomic DNA from this strain and cloned into the XhoI/NotI sites of pRS304. pMic60-EGFP and Rim1-GFP integrate at the *ura3-1* and *leu2-3* loci, respectively, and were introduced by linearization (using ApaI for pMic60-EGFP and EcoRV for Rim1-GFP) and lithium acetate transformation, followed by selection on SD media. Expression was verified by fluorescence microscopy.

## Yeast growth assays

Cells were grown to exponential phase in YPD, pelleted, and resuspended in water at a concentration of 0.5 OD$_{600}$/ml. 5 µl of 10-fold serial dilutions were plated on YPD and YPEG plates and cells were grown for ~36 hr (YPD) and ~48 hr (YPEG) at 30°C.

## Fluorescence microscopy

For all fluorescence microscopy experiments, cells were grown to exponential phase in SD with appropriate plasmid selection, concentrated, and spread on a 3% low-melt agarose pad set on concave microscope slides.

Images in *Figure 2* were captured with a DeltaVision Real Time (GE Healthcare Life Sciences; Picastaway, NJ) microscope using a 60×, 1.4 NA objective and a CoolSnap HQ camera (Photometrics; Tuscon, AZ). Z-series were taken with a 0.2 µm step size. Images were deconvolved using SoftWoRx software (GE Healthcare Life Sciences), and ImageJ (NIH; Bethesda, MD) was used to make linear adjustments to images.

For all other fluorescence images, including quantification of morphology and foci, Z-series of cells using a 0.2 µm step size were collected using the spinning-disk module of a Marianas SDC Real Time 3D Confocoal-TIRF microscope (Intelligent Imaging Innovations; Denver, CO) fitted with a 100×, 1.46 NA objective and either a Photometrics QuantEM EMCCD camera or a Hamamatsu (Japan) Orca Flash 4.0 sCMOS camera. Images were captured with SlideBook (Intelligent Imaging Innovations) and linear adjustments were made using ImageJ. Morphological and foci counting analysis was performed

manually using ImageJ and for each experiment, at least 75 cells from at least three fields of view were quantified.

To quantify the localization of Mic27 foci relative to Qcr2 in cells (*Figure 4D–F* and *Figure 4—figure supplement 4*), cells where Qcr2-mCherry was localized to mitochondrial subregions when compared to mito-TagBFP were chosen blind to Mic27 signal. Single plane images were compared to maximum z-projections to confirm the z-projection was representative of cell appearance. Qcr2 signal was considered positive when it was above an arbitrary threshold using the 'Moments' algorithm of ImageJ. Tracings of the mitochondrial perimeter and subregions of the perimeter adjacent to Qcr2 were manually performed. After identifying Qcr2-positive mitochondrial subregions, Mic27 foci were identified blind to Qcr2 signal and then compared to the Qcr2-positive regions of the tracings. Finally, Mic27 foci were scored as adjacent or not to Qcr2 signal in the z-projection and verified in the single plane images.

## Electron microscopy

Preparation of cells for morphology was performed as described previously described (*Rieder et al., 1996*). Briefly, cells were pelleted and fixed in 3% glutaraldehyde contained in 0.1 M sodium cacodylate, pH 7.4, 5 mM $CaCl_2$, 5 mM $MgCl_2$, and 2.5% sucrose for 1 hr at 22°C with gentle agitation, spheroplasted, embedded in 2% ultra low temperature agarose (prepared in water), cooled, and subsequently cut into small pieces (1 $mm^3$). The cells were then postfixed in 1% $OsO_4$/1% potassium ferrocyanide contained in 0.1 M cacodylate/5 mM $CaCl_2$, pH 7.4, for 30 min at 22°C. The blocks were washed thoroughly 4 times with double-distilled $H_2O$ ($ddH_2O$; 10 min in total), transferred to 1% thiocarbohydrazide at 22°C for 3 min, washed in $ddH_2O$ (4 times for 1 min each), and transferred to 1% $OsO_4$/1% potassium ferrocyanide in cacodylate buffer, pH 7.4, for an additional 3 min at 25°C. The cells were washed 4 times with $ddH_2O$ (15 min in total), en bloc stained in Kellenberger's uranyl acetate for 2 hr to overnight, dehydrated through a graded series of ethanol, and subsequently embedded in Spurr resin. Sections were cut on an ultramicrotome (Ultracut T; Reichert), poststained with uranyl acetate and lead citrate, and observed on a transmission electron microscope (Tecnai 12;FEI; Hillsboro, OR) at 100 kV. Images were recorded with a digital camera (Soft Imaging System MegaView III; Olympus; Japan), and figures were assembled in Photoshop (Adobe; San Jose, CA) with only linear adjustments in contrast and brightness.

## Whole cell extracts

Cells were grown to exponential phase in YPD, and whole cell lysates of 0.25 $OD_{600}$ cells were obtained by alkaline extraction (0.255 M NaOH, 1% beta-mercaptoethanol) followed by precipitation in 9% trichloroacetic acid. Precipitates were washed in acetone, dried, and resuspended in MURB protein sample buffer (100 mM MES, pH 7.0, 1% SDS, 3 M urea, 10% β-mercaptoethanol).

## Immunopurification and proteomic analysis

Immunopurifications were performed as previously described (*Hoppins et al., 2011*), except 1000 $OD_{600}$ cells were used. Briefly, cells were grown to exponential phase in YPD, resuspended in lysis buffer (20 mM HEPES pH7.4, 150 mM KOAc, 2 mM Mg(Ac)$_2$, 1 mM EGTA, 0.6 M sorbitol, and 1× Protease Inhibitor Mixture I [EMD Millipore; Billerica, MA]), flash-frozen dropwise in liquid $N_2$, and lysed using a Freezer/Mill (SPEX; Metuchen, MJ). As described, the cell lysate was cleared, cross-linked for 30 min with 1 mM DSP (Thermo Fisher Scientific; Waltham, MA), solubilized with 1% digitonin for 30 min, and pelleted again. The resulting supernatant was used for purifications with either 3 μg α-FLAG antibody (1: 1000, Sigma–Aldrich; St. Louis, MO) and 50 μl μMACS protein G beads (Miltenyi Biotec; San Diego, CA), or with 50 μl μMACS α-GFP MicroBeads (Miltenyi Biotec), and beads were isolated with μ columns and a μMACS separator (Miltenyi Biotec). For Western blot analysis, samples were eluted from beads with MURB sample buffer. For mass spectrometry proteomic analysis, samples were eluted using on-bead trypsin digestion as described previously and submitted to the University of California, Davis, Genome Center Proteomics Core. Sample processing and LC-MS/MS analysis were performed as previously described (*Hoppins et al., 2011*). Raw data from each mass spectrometry run are shown in source data files. For comparative analysis of MICOS interactions in different strain backgrounds, we normalized using the relative number of total immunopurified spectral counts in the mutant extracts to wild type extracts. All graphs of proteomic analyses display the range of values from duplicate experiments.

## Mic60 solubility analysis

Mitochondria from wild type or ΔMICOS cells expressing Mic60-EGFP tagged at its endogenous locus were isolated by differential centrifugation as described previously (*Hoppins et al., 2011*).

Mitochondria were resuspended in lysis buffer (20 mM HEPES pH7.4, 0.6 M sorbitol, 1 mM DTT, 0.1 M NaCl, 1× Protease Inhibitor Mixture I, and 1% Triton X-100) and incubated on ice for 30 min, followed by centrifugation at 50,000×$g$ for 60 min (TLA-100, Beckman Coulter; Brea, CA). Total, supernatant, and pellet fractions were resuspended in equivalent volumes of MURB sample buffer, and analyzed by Western blotting.

### Western blot analysis

Samples were boiled for 5 min and analyzed by SDS-PAGE, transferred to PVDF or nitrocellulose, and immunoblotted with the following primary antibodies at the indicated concentrations: mouse α-FLAG (1:1000, Sigma–Aldrich); mouse α-GFP (1:2000, UC Davis NeuroMab clone N86/8) or (1:2000, Thermo Fisher Scientific clone GF28R); rabbit α-mCherry (1:2000, Thermo Fisher Scientific); rabbit α-G6PDH (1:2000, Sigma–Aldrich); mouse α-Porin (1:1000, Thermo Fisher Scientific). The appropriate secondary antibodies conjugated to DyLight 680 and DyLight 800 (1:10000, Thermo Fisher Scientific) were used and visualized with the Odyssey Infrared Imaging System (LI-COR; Lincoln, NE). Linear adjustments to images were made using Adobe Photoshop.

### BN-PAGE analysis, in gel activity assays, and two-dimensional gel anaylsis

Crude mitochondrial extracts were isolated by differential centrifugations as described previously (*Hoppins et al., 2011*). Blue native electrophoresis (BN-PAGE) was performed in 4–16% gradient gels according to the recommendation of the Novex NativePAGE Bis-Tris gel System (Thermo Fisher Scientific). Briefly, 100 µg of isolated mitochondria was solubilized with 6 g digitonin per g mitochondrial protein. The extracts were centrifuged at 4°C for 15 min at 20,000×$g$, and aliquots of the supernatant (20 µl) were immediately loaded on the top of a 4–16% polyacrylamide gel. After electrophoresis, the gel was divided into strips, which were incubated in different solutions at room temperature for 20 min to 1 hr to reveal in gel activity. In order to reveal in gel ATPase activity, gel strips were incubated in a solution of 5 mM ATP, 5 mM MgCl$_2$, 0.05% lead acetate, 50 mM glycine–NaOH pH 8.4. For the cytochrome-$c$ oxidase activity, strips were incubated in the following solution (diaminobenzidine 0.6% [wt/vol], bovine heart cytochrome $c$ 1.2% [wt/vol], 1 nM catalase, 50 mM H$_2$PO$_4$, pH:7).

The Blue Native/SDS-PAGE two-dimensional analysis was performed as previously described (*Wittig et al., 2006*). Briefly, BN-PAGE was first performed as above except mitochondria were solubilized with 1 g n-Dodecyl β-D-maltoside (DDM) per g mitochondrial protein. After BN-PAGE, the gel was divided into strips which were incubated in 1% SDS buffer for 2 hr. The strips were then placed on top of a 16% Tricine-SDS polyacrylamide gel (*Schagger, 2006*). After electrophoresis in the second dimension, the gel was stained by classical silver staining.

## Acknowledgements

We thank members of the Nunnari lab for discussions and critical reading of the manuscript. We also thank Dr Michael Paddy in the MCB Imaging Facility at UC Davis for assistance with microscopy and Dr Brett Phinney and Darren Weber at the proteomics core facility at UC Davis for help with proteomic analyses. JN is supported by NIH grants R01GM062942, R01GM097432 and R01GM106019. JF is supported by a fellowship from the Jane Coffin Childs Memorial Fund for Medical Research.

## Additional information

#### Competing interests

JN: Reviewing editor, *eLife*. On Scientific Advisory Board of Mitobridge, and declares no financial interest related to this work. The other authors declare that no competing interests exist.

#### Funding

| Funder | Grant reference | Author |
| --- | --- | --- |
| National Institutes of Health (NIH) | R01GM062942 | Jodi Nunnari |
| Jane Coffin Childs Memorial Fund for Medical Research | fellowship | Jonathan R Friedman |

| Funder | Grant reference | Author |
| --- | --- | --- |
| National Institutes of Health (NIH) | R01GM097432 | Jodi Nunnari |
| National Institutes of Health (NIH) | R01GM106019 | Jodi Nunnari |

The funders had no role in study design, data collection and interpretation, or the decision to submit the work for publication.

## Author contributions

JRF, AM, Conception and design, Acquisition of data, Analysis and interpretation of data, Drafting or revising the article, Contributed unpublished essential data or reagents; JY, Acquisition of data, Drafting or revising the article; JMMC, Acquisition of data, Analysis and interpretation of data, Drafting or revising the article, Contributed unpublished essential data or reagents; JN, Conception and design, Analysis and interpretation of data, Drafting or revising the article

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
