## [Decision Letter]

Thank you for choosing to send your work entitled “MICOS, respiratory complexes and mitochondrial lipids coordinately build a functional mitochondrial inner membrane” for consideration at *eLife*. Your full submission has been evaluated by Vivek Malhotra (Senior editor) and two peer reviewers, one of whom, Richard Youle, is a member of our Board of Reviewing Editors, and the decision was reached after discussions between the reviewers and editors. Based on our discussions, we find that your manuscript, albeit of considerable interest, cannot be accepted for publication in its current form. You will need to provide new data to support your proposal, but if you are able to address the concerns in a timely manner we would request the same reviewers for their recommendation.

Four major issues were raised by the reviewers that would need to be resolved experimentally:

1) The work would benefit from more EM or super resolution analyses of MICOS protein localizations relative to predicted cristae junctions. Currently, protein localization assessments are indirect (near Ox Phos complex proteins in Figure 7) and of low resolution. An important conclusion is that the Mic27/Mic10/Mic12 complex localizes to cristae junctions independent of Mic60. This is based on the correlated locations of Mic27 and respiratory complexes presented in Figure 4. At least a quantification (which is admittedly a challenge on its own) of the proximity of Mic27 to Qcr2 and Atp2 and addressing whether they are more adjacent to each other than to some other mitochondrial substructure (nucleoids?) is needed to support this conclusion. This would assure that not everything is close together within mitochondria visualized at this low resolution. However, immune-Ems would be the best solution.

2) One related issue is the identification of MICOS “subcomplexes” found when subsets or individual components of MICOS are expressed in delta MICOS cells. Could the Mic60 foci in Figure 2 be misfolded aggregates? Perhaps expression of Mic19 stabilizes Mic60 allowing it to fold more normally. Immuno-EM may allow the authors to support the claim that Mic60 labels discrete structures (cristae junctions?) rather than forming nonspecific aggregates. Or perhaps blue native gels would support the contention that Mic60 forms discrete substructures. It would be important to back up or substantiate the authors’ conclusion at the end of the subsection headed “Mic10, Mic12, and Mic27 form a second independent MICOS organizing center” that Mic60 “has an intrinsic self-organizational capacity” above and beyond self-aggregation. Self-aggregation may also explain why Mic60 foci form in Rho0 cells. On the other hand, Figure 7 shows that Mic60 forms focal assemblies in Mic19 null cells that overlap with Mic27. This supports the authors’ idea that Mic60 is localized to actual substructures but in Figure 7 (relative to Figure 2) location of Mic60 may be influenced by other MICOS proteins and prevent aggregation.

3) The authors describe that alterations in cardiolipin affect Mic27/10/12 assembly, but not the Mic60/19 sub complex, which is consistent with the previous notion that mammalian Mic27 binds cardiolipin. However, loss of cardiolipin leads to disassembly of respiratory chain super complexes, which have been linked to cristae morphogenesis. It remains unclear how the authors distinguish between a direct effect of cardiolipin on Mic27 from that caused by an assembly defect of respiratory chain super complexes (even more as they observe that impaired assembly of respiratory complexes affects MICOS sub complexes).

4) Based on in-gel assays of respiratory chain complexes the authors propose that the loss of MICOS complexes does not affect assembly, but localization/positioning of respiratory complexes and thus impair their activity. This is highly speculative. Is it possible that complex IV lacks accessory factor(s) in MICOS-deficient cells? The authors should at least perform EM analysis to examine whether deletions of CBS1 or MSS51 restore cristae in the absence of Mic60 or MICOS.

Two more minor issues were also raised that would be valuable to address:

A) The point made in the Abstract that MICOS subunits have non-redundant functions has already been reported by the authors and others (Dev. Cell 21:694, 2011; FEBS J. 280:4943, 2013). For thoroughness, these prior papers describing non-redundant import related functions of Mic60 should be cited to high light the non-redundant genetic interactions (Hoppins 2011).

B) The dependence of Mic27/10/12 assemblies on mitochondrial DNA and their functional link to respiratory complexes is very intriguing. However, it is unclear how MICOS complexes affect mtDNA and vice versa. Restoration of cristae structures upon loss of the ATP synthase (20) or upon the loss of mtDNA (21) has been described and the reported findings do not provide significant new insight into the molecular basis of these observations.

---

## [Author Response]

*1) The work would benefit from more EM or super resolution analyses of MICOS protein localizations relative to predicted cristae junctions. Currently, protein localization assessments are indirect (near Ox Phos complex proteins in*
Figure 7*) and of low resolution. An important conclusion is that the Mic27/Mic10/Mic12 complex localizes to cristae junctions independent of Mic60. This is based on the correlated locations of Mic27 and respiratory complexes presented in*
Figure 4*. At least a quantification (which is admittedly a challenge on its own) of the proximity of Mic27 to Qcr2 and Atp2 and addressing whether they are more adjacent to each other than to some other mitochondrial substructure (nucleoids?) is needed to support this conclusion. This would assure that not everything is close together within mitochondria visualized at this low resolution. However, immune-Ems would be the best solution*.

We agree that one important point suggested by our data is that both Mic60 and Mic27/10/12 subcomplexes independently localize to cristae junctions. Data supporting this conclusion include the dependence of Mic27 foci formation on the presence of cristae as well as the adjacent localization of Mic27 foci to cristae-localized respiratory complex proteins *∆mic60* cells. We agree that more equivocal support would include immuno-EM data demonstrating more directly the localization of Mic27 to cristae junctions. However, immuno-EM analysis of Mic27 in *∆mic60* cells, as mentioned in the manuscript, proved too technically challenging because of the low abundance of both cristae junctions and Mic27 foci in *∆mic60* cells and the difficulty associated with visualizing cristae junctions in thin section. However, in order to determine if this problem was specific to *∆mic60* cells, we performed immuno-labeling of Mic60 in both *∆mic19* cells and ∆MICOS cells. We obtained images of gold particle clusters specific for Mic60 at the edges of mitochondria (as observed for Mic27 immuno-gold labeling), which likely correspond to the focal appearance we observe by fluorescence. However as seen previously for Mic27, we could not determine whether these were spatially linked to cristae junctions because of the morphology of the mitochondrial membranes in these thin sections.

Given the importance of this conclusion, however, we further analyzed our light imaging data by quantitatively assessing the proximity of Mic27 and Mic60 foci to substructures labeled by Qcr2 and Atp2, proteins previously demonstrated to be enriched in cristae. Per the reviewers’ request, we acquired higher resolution images using a CMOS camera with ∼2x greater pixel density (see new Figure 4, Figure 4–figure supplement 5-6, Figure 7, and Figure 7–figure supplement 3). We think these images convincingly demonstrate the localization of both Mic27 and Mic60 foci immediately adjacent to submitochondrial-concentrated regions of Qcr2 and Atp2. Thus, to further test this, we developed an unbiased quantitative analysis method of these higher resolution images, which indicates that the localization pattern of Mic27 foci adjacent to Qcr2-labled cristae is non-random (see new Figure 4 and Figure 4—figure supplement 5).

*2) One related issue is the identification of MICOS* “*subcomplexes*” *found when subsets or individual components of MICOS are expressed in delta MICOS cells. Could the Mic60 foci in*
Figure 2
*be misfolded aggregates? Perhaps expression of Mic19 stabilizes Mic60 allowing it to fold more normally. Immuno-EM may allow the authors to support the claim that Mic60 labels discrete structures (cristae junctions?) rather than forming nonspecific aggregates. Or perhaps blue native gels would support the contention that Mic60 forms discrete substructures. It would be important to back up or substantiate the authors’ conclusion at the end of the subsection headed “Mic10, Mic12, and Mic27 form a second independent MICOS organizing center” that Mic60* “*has an intrinsic self-organizational capacity*” *above and beyond self-aggregation. Self-aggregation may also explain why Mic60 foci form in Rho0 cells. On the other hand,*
Figure 7
*shows that Mic60 forms focal assemblies in Mic19 null cells that overlap with Mic27. This supports the authors’ idea that Mic60 is localized to actual substructures but in*
Figure 7
*(relative to*
Figure 2*) location of Mic60 may be influenced by other MICOS proteins and prevent aggregation*.

To address the concern that Mic60 foci represent misfolded aggregateswe simply tested whether Mic60 behaves as a soluble protein upon detergent solubilization of isolated mitochondria extracts under low salt conditions (see new Figure 2), To address the concern that Mic60 foci represent misfolded aggregates, we simply tested whether Mic60 behaves as a soluble protein upon detergent solubilization of isolated mitochondria extracts under low salt conditions (see new Figure 2). Specifically, we solubilized mitochondria from wild type and ∆MICOS cells expressing Mic60-EGFP using 100mM NaCl and 1% TX-100 and performed a 50,000g spin for 1h, which is sufficient to pellet protein aggregates 60S or greater (equivalent to ∼2 MDa 60S ribosome). Under these conditions, Fcj1-GFP in both the wild type (sized at 0.6 to 1.5 MDa in published work) and in ∆MICOS mitochondrial extracts was quantitatively recovered in the supernatant fraction. These data indicate that Mic60-EGFP foci in ∆MICOS cells do not represent non-specific aggregates, and are likely self-assemblies or form as a consequence of another determinant.

*3) The authors describe that alterations in cardiolipin affect Mic27/10/12 assembly, but not the Mic60/19 sub complex, which is consistent with the previous notion that mammalian Mic27 binds cardiolipin. However, loss of cardiolipin leads to disassembly of respiratory chain super complexes, which have been linked to cristae morphogenesis. It remains unclear how the authors distinguish between a direct effect of cardiolipin on Mic27 from that caused by an assembly defect of respiratory chain super complexes (even more as they observe that impaired assembly of respiratory complexes affects MICOS sub complexes)*.

In our model (Figure 7)we include the possibility that the effect of cardiolipin on respiratory supercomplex assembly may indirectly influence Mic27 subcomplex assembly, In our model (Figure 7), we include the possibility that the effect of cardiolipin on respiratory supercomplex assembly may indirectly influence Mic27 subcomplex assembly. However, while decreases in cardiolipin are linked to disassembly of respiratory chain super complexes, to date there is no link between cristae morphogenesis and supercomplex formation in yeast cells, with the exception of ATP synthase dimerization, which is not altered by loss of cardiolipin in cells. However, it is possible that cardiolipin synthesis may indirectly regulate Mic27 subcomplex assembly through an effect on respiratory complex assembly, but not through defects in cristae morphogenesis. An equally probable possibility suggested by our data is that Mic27 subcomplexes may directly sense cardiolipin through its apolipoprotein-like domain. We have made sure not to overstate this conclusion.

*4) Based on in-gel assays of respiratory chain complexes the authors propose that the loss of MICOS complexes does not affect assembly, but localization/positioning of respiratory complexes and thus impair their activity. This is highly speculative. Is it possible that complex IV lacks accessory factor(s) in MICOS-deficient cells? The authors should at least perform EM analysis to examine whether deletions of CBS1 or MSS51 restore cristae in the absence of Mic60 or MICOS*.

Our data do not definitively determine the cause of the respiratory defect in the absence of MICOS, Our data do not definitively determine the cause of the respiratory defect in the absence of MICOS, beyond noting that there is a subtle defect in complex IV activity. However, our analysis also indicates that respiratory complexes and supercomplexes are intact in ∆MICOS cells. In order to test the status of respiratory complex composition more thoroughly, we performed 2D native SDS-PAGE gel analysis of detergent-solubilized purified mitochondria (see new Figure 6). These data show no apparent defects in the composition of assembled respiratory complexes in either ∆mic60 or ∆MICOS cells, indicating that ∆MICOS cells do not lack complex IV accessory factor(s). We are currently delving into the cause of the respiratory deficiency in ∆MICOS cells by examining suppressors, but we think that this is beyond the scope of the current study.

The reviewers also suggest that we perform EM analysis on cells deficient for Complex III or IV assembly factors to test if cristae are restored in ∆MICOS cells in their absence. Based on this suggestion, the reviewers likely misunderstood our data from ∆MICOS rho0 cells. Specifically, in ∆MICOS cells only aberrant mitochondrial morphology is restored upon loss of respiratory complexes (rho0), not cristae structure (Figure 6). Indeed, restoration of morphology correlates with loss of cristae in rho0 cells, which indicates that the aberrant cristae, observed by light imaging of Qcr2, causes mitochondrial morphological defects in ∆MICOS, a finding we previously published for single MICOS subunit deletion cells (Hoppins et al. JCB 2011). Thus, while deletion of CBS1 and MSS51 partially restore tubular morphology in ∆MICOS cells (Figure 6), this is not mediated through restoration of cristae morphology.

*Two more minor issues were also raised that would be valuable to address*:

*A) The point made in the Abstract that MICOS subunits have non-redundant functions has already been reported by the authors and others (Dev. Cell 21:694, 2011; FEBS J. 280:4943, 2013). For thoroughness, these prior papers describing non-redundant import related functions of Mic60 should be cited to high light the non-redundant genetic interactions (Hoppins 2011)*.

We thank the reviewer for this suggestion and have now specifically highlighted the MICOS-independent import-related function of Mic60 in the Introduction. While the FEBS J reference the reviewer suggested shows a Mic60-specific role in accumulation of mutant Sod1, this is not necessarily an import-related function of Mic60, and thus we do not cite it here.

*B) The dependence of Mic27/10/12 assemblies on mitochondrial DNA and their functional link to respiratory complexes is very intriguing. However, it is unclear how MICOS complexes affect mtDNA and vice versa. Restoration of cristae structures upon loss of the ATP synthase (*[20]*) or upon the loss of mtDNA (*[21]*) has been described and the reported findings do not provide significant new insight into the molecular basis of these observations*.

As stated above for point 4, loss of ATP synthase (20) or loss of mtDNA ([20] and [21]) have been shown to alleviate mitochondrial morphology defects in the absence of MICOS components, however, neither work shows a restoration of normal cristae morphology. Our data indicate that mitochondrial DNA is maintained in ∆MICOS cells (though nucleoids often appear mis-distributed) (Figure 1) and correspondingly, respiratory complexes are still expressed and assembled in ∆MICOS cells (Figure 6). However, as stated for in our response to point 4 above, in ∆MICOS cells, deletion of respiratory complex assembly factors or loss of mtDNA causes loss of cristae, which is what is correlated with the suppression of mitochondrial morphological defects, not the restoration of cristae structure. The link between the presence of respiratory complexes and Mic27/10/12 assemblies but not Mic60 assemblies is a conceptual advance because it suggests that Mic60 marks nascent cristae sites and that Mic27/10/12 assemblies are recruited/assembled following cristae biogenesis driven by ATP synthase and respiratory complexes.